# ON MARGIN MAXIMIZATION IN LINEAR AND RELU NETWORKS

## ABSTRACT

The implicit bias of neural networks has been extensively studied in recent years. Lyu & Li (2019) showed that in homogeneous networks trained with the exponential or the logistic loss, gradient flow converges to a KKT point of the max margin problem in the parameter space. However, that leaves open the question of whether this point will generally be an actual optimum of the max margin problem. In this paper, we study this question in detail, for several neural network architectures involving linear and ReLU activations. Perhaps surprisingly, we show that in many cases, the KKT point is not even a *local* optimum of the max margin problem. On the flip side, we identify multiple settings where a local or global optimum can be guaranteed. Finally, we answer a question posed in Lyu & Li (2019) by showing that for *non-homogeneous* networks, the normalized margin may strictly decrease over time.

## 1  INTRODUCTION

A central question in the theory of deep learning is how neural networks generalize even when trained without any explicit regularization, and when there are far more learnable parameters than training examples. In such optimization problems there are many solutions that label the training data correctly, and gradient descent seems to prefer solutions that generalize well (Zhang et al., 2016). Hence, it is believed that gradient descent induces an *implicit bias* (Neyshabur et al., 2014; 2017), and characterizing this bias has been a subject of extensive research in recent years.

A main focus in the theoretical study of implicit bias is on *homogeneous* neural networks. These are networks where scaling the parameters by any factor $\alpha > 0$ scales the predictions by $\alpha^L$ for some constant $L$. For example, fully-connected and convolutional ReLU networks without bias terms are homogeneous. Lyu & Li (2019) proved that in linear and ReLU homogeneous networks trained with the exponential or the logistic loss, if gradient flow converges to zero loss[1], then the direction to which the parameters of the network converge can be characterized as a first order stationary point (KKT point) of the maximum margin problem in the parameter space. Namely, the problem of minimizing the $\ell_2$ norm of the parameters under the constraints that each training example is classified correctly with margin at least $1$. They also showed that this KKT point satisfies necessary conditions for optimality. However, the conditions are not known to be sufficient even for local optimality. It is analogous to showing that some unconstrained optimization problem converges to a point with gradient zero, without proving that it is either a global or a local minimum.

In this work we consider several architectures of homogeneous neural networks with linear and ReLU activations, and study whether the aforementioned KKT point is guaranteed to be a global optimum of the maximum margin problem, a local optimum, or neither. Perhaps surprisingly, our results imply that in many cases, such as depth-2 fully-connected ReLU networks and depth-2 diagonal linear networks, the KKT point may not even be a *local* optimum of the maximum-margin problem. On the flip side, we identify multiple settings where a local or global optimum can be guaranteed.

We now describe our results in a bit more detail. We denote by $\mathcal{N}$ the class of neural networks without bias terms, where the weights in each layer might have an arbitrary sparsity pattern, and

---

[1] They also assumed directional convergence, but (Ji & Telgarsky, 2020) later showed that this assumption is not required.

Table 1: Results on depth-2 networks.

| | Linear | ReLU |
|---|---|---|
| Fully-connected | Global (Thm. 3.1) | Not local (Thm. 3.2) |
| $\mathcal{N}_{\text{no-share}}$ | Not local (Thm. 4.1) | Not local (Thm. 3.2) |
| $\mathcal{N}_{\text{no-share}}$ assuming non-zero weights vectors | Global (Thm. 4.2) | Not local (Thm. 4.2) |
| $\mathcal{N}_{\text{no-share}}$ assuming non-zero inputs to all neurons | Global (Thm. 4.2) | Local, Not global (Thm. 4.3) |
| $\mathcal{N}$ assuming non-zero inputs to all neurons | Not local (Thm. 4.4) | Not local (Thm. 4.4) |

Table 2: Results on deep networks.

| | Linear | ReLU |
|---|---|---|
| Fully-connected | Global (Thm. 3.1) | Not local (Thm. 3.2) |
| $\mathcal{N}_{\text{no-share}}$ assuming non-zero inputs to all neurons | Not local (Thm. 5.1) | Not local (Thm. 5.1) |
| $\mathcal{N}$ - max margin for each layer separately | Global (Thm. 5.2) | Not local (Thm. 5.3) |
| $\mathcal{N}$ - max margin for each layer separately, assuming non-zero inputs to all neurons | Global (Thm. 5.2) | Local, Not global (Thm. 5.4) |

weights might be shared[2]. The class $\mathcal{N}$ contains, for example, *convolutional networks*. Moreover, we denote by $\mathcal{N}_{\text{no-share}}$ the subclass of $\mathcal{N}$ that contains only networks without shared weights, such as *fully-connected networks* and *diagonal networks* (cf. Gunasekar et al. (2018b); Yun et al. (2020)). We describe our main results below, and also summarize them in Tables 1 and 2.

**Fully-connected networks:**

- In linear fully-connected networks of any depth the KKT point is a global optimum[3].

- In fully-connected depth-2 ReLU networks the KKT point may not even be a local optimum. Moreover, this negative result holds with constant probability over the initialization, i.e., there is a training dataset such that gradient flow with random initialization converges with constant probability to the direction of a KKT point which is not a local optimum.

**Depth-$2$ networks in $\mathcal{N}$:**

- The positive result on fully-connected linear networks does not extend to networks with sparse weights: In linear diagonal networks the KKT point may not be a local optimum.

- In our proof for the above negative result, the KKT point contains a neuron whose weights vector is zero. However, in practice gradient descent often converges to networks that do not contain such zero neurons. We show that for linear networks in $\mathcal{N}_{\text{no-share}}$, if the KKT point has only non-zero weights vectors, then it is a global optimum. We also show that even for the simple case of depth-2 diagonal linear networks, the optimality of the KKT points can be unexpectedly subtle, in the context of margin maximization in the predictor space (see Remark 4.1).

- For ReLU networks in $\mathcal{N}_{\text{no-share}}$, in order to obtain a positive result we need a stronger assumption. We show that if the KKT point is such that for every input in the dataset the input to every hidden neuron in the network is non-zero, then it is guaranteed to be a local optimum (but not necessarily a global optimum).

- For linear or ReLU convolutional networks, even if the above assumptions hold, the KKT point may not be a local optimum.

**Deep networks in $\mathcal{N}$:**

- We show that the positive results on depth-2 linear and ReLU networks in $\mathcal{N}_{\text{no-share}}$ (under the assumptions described above) do not apply to deeper networks.

- We study a weaker notion of margin maximization: maximizing the margin for each layer separately. For linear networks of depth $m \geq 2$ in $\mathcal{N}$ (including networks with shared weights), we show that the KKT point is a global optimum of the per-layer maximum margin problem. For ReLU networks the KKT point may not even be a local optimum of this problem, but under the assumption on non-zero inputs to all neurons it is a local optimum.

---

[2]See Section 2 for the formal definition.

[3]We note that margin maximization for such networks in the predictor space is already known (Ji & Telgarsky, 2020). However, margin maximization in the predictor space does not necessarily imply margin maximization in the parameter space.

In the paper, our focus is on understanding what can be guaranteed for the KKT convergence points specified in Lyu & Li (2019). Accordingly, in most of our negative results, the construction assumes some specific initialization of gradient flow, and does not quantify how "likely" they are to be reached under some random initialization. An exception is our negative result for depth-2 fully-connected ReLU networks (Thm. 3.2), which holds with constant probability under reasonable random initializations. Understanding whether this can be extended to the other settings we consider is an interesting problem for future research.

Finally, we consider *non-homogeneous networks*, for example, networks with skip connections or bias terms. Lyu & Li (2019) showed that a smoothed version of the *normalized margin* is monotonically increasing when training homogeneous networks. They observed empirically that the normalized margin is monotonically increasing also when training non-homogeneous networks, but did not provide a proof for this phenomenon and left it as an open problem. We give an example for a simple non-homogeneous network where the normalized margin (as well as the smoothed margin) is strictly *decreasing* (see Thm. 6.1).

The paper is structured as follows: In Section 2 we provide necessary notations and definitions, and discuss the most relevant prior results. Additional related works are discussed in Appendix A. In Sections 3, 4 and 5 we state our results on fully-connected networks, depth-2 networks in $\mathcal{N}$ and deep networks in $\mathcal{N}$ respectively, and provide some proof ideas. In Section 6 we state our result on non-homogeneous networks. All formal proofs are deferred to Appendix C.

## 2 PRELIMINARIES

**Notations.** We use bold-faced letters to denote vectors, e.g., $\mathbf{x} = (x_1, \ldots, x_d)$. For $\mathbf{x} \in \mathbb{R}^d$ we denote by $\|\mathbf{x}\|$ the Euclidean norm. We denote by $\mathbb{1}(\cdot)$ the indicator function, for example $\mathbb{1}(t \geq 5)$ equals 1 if $t \geq 5$ and 0 otherwise. For an integer $d \geq 1$ we denote $[d] = \{1, \ldots, d\}$.

**Neural networks.** A *fully-connected neural network* $\Phi$ of depth $m \geq 2$ is parameterized by a collection $\boldsymbol{\theta} = [W^{(l)}]_{l=1}^m$ of weight matrices, such that for every layer $l \in [m]$ we have $W^{(l)} \in \mathbb{R}^{d_l \times d_{l-1}}$. Thus, $d_l$ denotes the number of neurons in the $l$-th layer (i.e., the *width* of the layer). We assume that $d_m = 1$ and denote by $d := d_0$ the input dimension. The neurons in layers $[m-1]$ are called *hidden neurons*. A fully-connected network computes a function $\Phi(\boldsymbol{\theta}; \cdot) : \mathbb{R}^d \to \mathbb{R}$ defined recursively as follows. For an input $\mathbf{x} \in \mathbb{R}^d$ we set $\mathbf{h}_0' = \mathbf{x}$, and define for every $j \in [m-1]$ the input to the $j$-th layer as $\mathbf{h}_j = W^{(j)} \mathbf{h}_{j-1}'$, and the output of the $j$-th layer as $\mathbf{h}_j' = \sigma(\mathbf{h}_j)$, where $\sigma : \mathbb{R} \to \mathbb{R}$ is an activation function that acts coordinate-wise on vectors. Then, we define $\Phi(\boldsymbol{\theta}; \mathbf{x}) = W^{(m)} \mathbf{h}_{m-1}'$. Thus, there is no activation function in the output neuron. When considering depth-2 fully-connected networks we often use a parameterization $\boldsymbol{\theta} = [\mathbf{w}_1, \ldots, \mathbf{w}_k, \mathbf{v}]$ where $\mathbf{w}_1, \ldots, \mathbf{w}_k$ are the weights vectors of the $k$ hidden neurons (i.e., correspond to the rows of the first layer's weight matrix) and $\mathbf{v}$ are the weights of the second layer.

We also consider neural networks where some weights can be missing or shared. We define a class $\mathcal{N}$ of networks that may contain sparse and shared weights as follows. A network $\Phi$ in $\mathcal{N}$ is parameterized by $\boldsymbol{\theta} = [\mathbf{u}^{(l)}]_{l=1}^m$ where $m$ is the depth of $\Phi$, and $\mathbf{u}^{(l)} \in \mathbb{R}^{p_l}$ are the parameters of the $l$-th layer. We denote by $W^{(l)} \in \mathbb{R}^{d_l \times d_{l-1}}$ the weight matrix of the $l$-th layer. The matrix $W^{(l)}$ is described by the vector $\mathbf{u}^{(l)}$, and a function $g_l : [d_l] \times [d_{l-1}] \to [p_l] \cup \{0\}$ as follows: $W_{ij}^{(l)} = 0$ if $g_l(i, j) = 0$, and $W_{ij}^{(l)} = u_k$ if $g_l(i, j) = k > 0$. Thus, the function $g_l$ represents the sparsity and weight-sharing pattern of the $l$-th layer, and the dimension $p_l$ of $\mathbf{u}^{(l)}$ is the number of free parameters in the layer. We denote by $d := d_0$ the input dimension of the network and assume that the output dimension $d_m$ is 1. The function $\Phi(\boldsymbol{\theta}; \cdot) : \mathbb{R}^d \to \mathbb{R}$ computed by the neural network is defined recursively by the weight matrices as in the case of fully-connected networks. For example, convolutional neural networks are in $\mathcal{N}$. Note that the networks in $\mathcal{N}$ do not have bias terms and do not allow weight sharing between different layers. Moreover, we define a subclass $\mathcal{N}_{\text{no-share}}$ of $\mathcal{N}$, that contains networks without shared weights. Formally, a network $\Phi$ is in $\mathcal{N}_{\text{no-share}}$ if for every layer $l$ and every $k \in [p_l]$ there is at most one $(i, j) \in [d_l] \times [d_{l-1}]$ such that $g_l(i, j) = k$. Thus, networks in $\mathcal{N}_{\text{no-share}}$ might have sparse weights, but do not allow shared weights. For example, diagonal networks (defined below) and fully-connected networks are in $\mathcal{N}_{\text{no-share}}$.

A *diagonal neural network* is a network in $\mathcal{N}_{\text{no-share}}$ such that the weight matrix of each layer is diagonal, except for the last layer. Thus, the network is parameterized by $\boldsymbol{\theta} = [\mathbf{w}_1, \ldots, \mathbf{w}_m]$ where $\mathbf{w}_j \in \mathbb{R}^d$ for all $j \in [m]$, and it computes a function $\Phi(\boldsymbol{\theta}; \cdot) : \mathbb{R}^d \to \mathbb{R}$ defined recursively as follows. For an input $\mathbf{x} \in \mathbb{R}^d$ set $\mathbf{h}_0 = \mathbf{x}$. For $j \in [m-1]$, the output of the $j$-th layer is $\mathbf{h}_j = \sigma(\text{diag}(\mathbf{w}_j)\mathbf{h}_{j-1})$. Then, we have $\Phi(\boldsymbol{\theta}; \mathbf{x}) = \mathbf{w}_m^\top \mathbf{h}_{m-1}$.

In all the above definitions the parameters $\boldsymbol{\theta}$ of the neural networks are given by a collection of matrices or vectors. We often view $\boldsymbol{\theta}$ as the vector obtained by concatenating the matrices or vectors in the collection. Thus, $\|\boldsymbol{\theta}\|$ denotes the $\ell_2$ norm of the vector $\boldsymbol{\theta}$.

The ReLU activation function is defined by $\sigma(z) = \max\{0, z\}$, and the linear activation is $\sigma(z) = z$. In this work we focus on ReLU networks (i.e., networks where all neurons have the ReLU activation) and on linear networks (where all neurons have the linear activation). We say that a network $\Phi$ is *homogeneous* if there exists $L > 0$ such that for every $\alpha > 0$ and $\boldsymbol{\theta}, \mathbf{x}$ we have $\Phi(\alpha\boldsymbol{\theta}; \mathbf{x}) = \alpha^L \Phi(\boldsymbol{\theta}; \mathbf{x})$. Note that in our definition of the class $\mathcal{N}$ we do not allow bias terms, and hence all linear and ReLU networks in $\mathcal{N}$ are homogeneous. With the exception of Section 6 which studies non-homogeneous networks, all networks considered in this work are homogeneous.

**Optimization problem and gradient flow (GF).** Let $S = \{(\mathbf{x}_i, y_i)\}_{i=1}^n \subseteq \mathbb{R}^d \times \{-1, 1\}$ be a binary classification training dataset. Let $\Phi$ be a neural network parameterized by $\boldsymbol{\theta} \in \mathbb{R}^m$. For a loss function $\ell : \mathbb{R} \to \mathbb{R}$ the empirical loss of $\Phi(\boldsymbol{\theta}; \cdot)$ on the dataset $S$ is $\mathcal{L}(\boldsymbol{\theta}) := \sum_{i=1}^n \ell(y_i \Phi(\boldsymbol{\theta}; \mathbf{x}_i))$. We focus on the exponential loss $\ell(q) = e^{-q}$ and the logistic loss $\ell(q) = \log(1 + e^{-q})$, and consider gradient flow (GF) on the objective $\mathcal{L}(\boldsymbol{\theta})$. This setting captures gradient descent with an infinitesimal step size. Let $\boldsymbol{\theta}(t)$ be the trajectory of GF. Starting from an initial point $\boldsymbol{\theta}(0)$, the dynamics of $\boldsymbol{\theta}(t)$ is given by the differential equation $\frac{d\boldsymbol{\theta}(t)}{dt} = -\nabla\mathcal{L}(\boldsymbol{\theta}(t))$. Note that the ReLU function is not differentiable at 0. Practical implementations of gradient methods define the derivative $\sigma'(0)$ to be some constant in $[0, 1]$. We note that the exact value of $\sigma'(0)$ has no effect on our results.

**Convergence to a KKT point of the maximum-margin problem.** We say that a trajectory $\boldsymbol{\theta}(t)$ *converges in direction* to $\tilde{\boldsymbol{\theta}}$ if $\lim_{t\to\infty} \frac{\boldsymbol{\theta}(t)}{\|\boldsymbol{\theta}(t)\|} = \frac{\tilde{\boldsymbol{\theta}}}{\|\tilde{\boldsymbol{\theta}}\|}$. In this work we rely on the following theorem:

**Theorem 2.1** (Paraphrased from Lyu & Li (2019); Ji & Telgarsky (2020)). *Let $\Phi$ be a homogeneous linear or ReLU neural network. Consider minimizing either the exponential or the logistic loss over a binary classification dataset $\{(\mathbf{x}_i, y_i)\}_{i=1}^n$ using GF. Assume that there exists time $t_0$ such that $\mathcal{L}(\boldsymbol{\theta}(t_0)) < 1$, namely, $\Phi$ classifies every $\mathbf{x}_i$ correctly. Then, GF converges in direction to a first order stationary point (KKT point) of the following maximum margin problem in parameter space:*

$$\min_{\boldsymbol{\theta}} \frac{1}{2}\|\boldsymbol{\theta}\|^2 \quad s.t. \quad \forall i \in [n] \ \ y_i\Phi(\boldsymbol{\theta}; \mathbf{x}_i) \geq 1 \ . \tag{1}$$

*Moreover, $\mathcal{L}(\boldsymbol{\theta}(t)) \to 0$ and $\|\boldsymbol{\theta}(t)\| \to \infty$ as $t \to \infty$.*

In the case of ReLU networks, Problem 1 is non-smooth. Hence, the KKT conditions are defined using Clarke's subdifferential, which is a generalization of the differential for non-differentiable functions. See Appendix B for a formal definition. We note that Lyu & Li (2019) proved the above theorem under the assumption that $\boldsymbol{\theta}$ converges in direction, and Ji & Telgarsky (2020) showed that such a directional convergence occurs and hence this assumption is not required.

Lyu & Li (2019) also showed that the KKT conditions of Problem 1 are necessary for optimality. In convex optimization problems, necessary KKT conditions are also sufficient for global optimality. However, the constraints in Problem 1 are highly non-convex. Moreover, the standard method for proving that necessary KKT conditions are sufficient for *local* optimality, is by showing that the KKT point satisfies certain *second order sufficient conditions (SOSC)* (cf. Ruszczynski (2011)). However, even when $\Phi$ is a linear neural network it is not known when such conditions hold. Thus, the KKT conditions of Problem 1 are not known to be sufficient even for local optimality.

A linear network with weight matrices $W^{(1)}, \ldots, W^{(m)}$ computes a linear predictor $\mathbf{x} \mapsto \langle \boldsymbol{\beta}, \mathbf{x} \rangle$ where $\boldsymbol{\beta} = W^{(m)} \cdot \ldots \cdot W^{(1)}$. Some prior works studied the implicit bias of linear networks in the *predictor space*. Namely, characterizing the vector $\boldsymbol{\beta}$ from the aforementioned linear predictor. Gunasekar et al. (2018b) studied the implications of margin maximization in the *parameter space* on the implicit bias in the predictor space. They showed that minimizing $\|\boldsymbol{\theta}\|$ (under the constraints

in Problem 1) implies: (1) Minimizing $\|\boldsymbol{\beta}\|_2$ for fully-connected networks; (2) Minimizing $\|\boldsymbol{\beta}\|_{2/L}$ for depth-$L$ diagonal networks; (3) Minimizing $\|\hat{\boldsymbol{\beta}}\|_{2/L}$ for depth-$L$ convolutional networks with full-dimensional filters, where $\hat{\boldsymbol{\beta}}$ are the Fourier coefficients of $\boldsymbol{\beta}$. However, these implications may not hold if GF converges to a KKT point which is not a global optimum of Problem 1.

For some classes of linear networks, positive results were obtained directly in the predictor space, without assuming convergence to a global optimum of Problem 1 in the parameter space. Most notably, for fully-connected linear networks (of any depth), Ji & Telgarsky (2020) showed that under the assumptions of Thm. 2.1, GF maximizes the $\ell_2$ margin in the predictor space. Note that margin maximization in the predictor space does not necessarily imply margin maximization in the parameter space. Moreover, some results on the implicit bias in the predictor space of linear convolutional networks with full-dimensional convolutional filters are given in Gunasekar et al. (2018b). However, the architecture and set of assumptions are different than what we focus on.

## 3 FULLY-CONNECTED NETWORKS

First, we show that fully-connected linear networks converge to a global optimum of Problem 1.

**Theorem 3.1.** *Let $m \geq 2$ and let $\Phi$ be a depth-$m$ fully-connected linear network parameterized by $\boldsymbol{\theta}$. Consider minimizing either the exponential or the logistic loss over a dataset $\{(\mathbf{x}_i, y_i)\}_{i=1}^n$ using GF. Assume that there exists time $t_0$ such that $\mathcal{L}(\boldsymbol{\theta}(t_0)) < 1$. Then, GF converges in direction to a global optimum of Problem 1.*

*Proof idea (for the complete proof see Appendix C.2).* Building on results from Ji & Telgarsky (2020) and Du et al. (2018), we show that GF converges in direction to a KKT point $\tilde{\boldsymbol{\theta}} = [\tilde{W}^{(1)}, \ldots, \tilde{W}^{(m)}]$ such that for every $l \in [m]$ we have $\tilde{W}^{(l)} = C \cdot \mathbf{v}_l \mathbf{v}_{l-1}^\top$, where $C > 0$ and $\mathbf{v}_0, \ldots, \mathbf{v}_m$ are unit vectors (with $\mathbf{v}_m = 1$). Also, we have $\|\tilde{W}^{(m)} \cdot \ldots \cdot \tilde{W}^{(1)}\| = C^m = \min \|\mathbf{u}\|$ s.t. $y_i \mathbf{u}^\top \mathbf{x}_i \geq 1$ for all $i \in [n]$. Then, we show that every $\boldsymbol{\theta}$ that satisfies these properties, and satisfies the constraints of Problem 1, is a global optimum. Intuitively, the most "efficient" way (in terms of minimizing the parameters) to achieve margin 1 with a linear fully-connected network, is by using a network such that the direction of its corresponding linear predictor maximizes the margin, the layers are balanced (i.e., have equal norms), and the weight matrices are aligned. $\square$

We now prove that the positive result in Thm. 3.1 does not apply to ReLU networks. We show that in depth-2 fully-connected ReLU networks GF might converge in direction to a KKT point of Problem 1 which is not even a local optimum. Moreover, it occurs under conditions holding with constant probability over reasonable random initializations.

**Theorem 3.2.** *Let $\Phi$ be a depth-$2$ fully-connected ReLU network with input dimension $2$ and two hidden neurons. Namely, for $\boldsymbol{\theta} = [\mathbf{w}_1, \mathbf{w}_2, \mathbf{v}]$ and $\mathbf{x} \in \mathbb{R}^2$ we have $\Phi(\boldsymbol{\theta}; \mathbf{x}) = \sum_{l=1}^2 v_l \sigma(\mathbf{w}_l^\top \mathbf{x})$. Consider minimizing either the exponential or the logistic loss using GF. Consider the dataset $\{(\mathbf{x}_1, y_1), (\mathbf{x}_2, y_2)\}$ where $\mathbf{x}_1 = \left(1, \frac{1}{4}\right)^\top$, $\mathbf{x}_2 = \left(-1, \frac{1}{4}\right)^\top$, and $y_1 = y_2 = 1$. Assume that the initialization $\boldsymbol{\theta}(0)$ is such that for every $i \in \{1, 2\}$ we have $\langle \mathbf{w}_1(0), \mathbf{x}_i \rangle > 0$ and $\langle \mathbf{w}_2(0), \mathbf{x}_i \rangle < 0$. Also, assume that $v_1(0) > 0$. Then, GF converges to zero loss, and converges in direction to a KKT point of Problem 1 which is not a local optimum.*

*Proof idea (for the complete proof see Appendix C.3).* By analyzing the dynamics of GF on the given dataset, we show that it converges to zero loss, and converges in direction to a KKT point $\tilde{\boldsymbol{\theta}}$ such that $\tilde{\mathbf{w}}_1 = (0, 2)^\top$, $\tilde{v}_1 = 2$, $\tilde{\mathbf{w}}_2 = \mathbf{0}$, and $\tilde{v}_2 = 0$. Note that $\tilde{\mathbf{w}}_2 = \mathbf{0}$ and $\tilde{v}_2 = 0$ since $\mathbf{w}_2(t), v_2(t)$ remain constant during the training and $\lim_{t \to \infty} \|\boldsymbol{\theta}(t)\| = \infty$. This is illustrated in the figure on the right: For time $t$ we denote $\mathbf{u}_i(t) = v_i(t) \mathbf{w}_i(t)$. We observe $\mathbf{u}_1(t), \mathbf{u}_2(t)$ for times $t_1 < t_2$. As $t \to \infty$ we have $\|\mathbf{u}_1(t)\| \to \infty$ and $\mathbf{u}_1$ converges in direction to $(0, 1)$. The vector $\mathbf{u}_2$ remains constant during the training. Hence $\frac{\mathbf{u}_2(t)}{\|\boldsymbol{\theta}(t)\|} \to \mathbf{0}$.

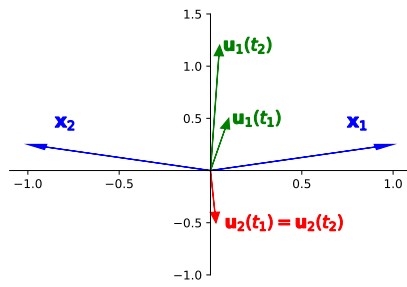

Then, we show that for every $0 < \epsilon < 1$ there exists some $\boldsymbol{\theta}'$ such that $\|\boldsymbol{\theta}' - \tilde{\boldsymbol{\theta}}\| \leq \epsilon$, $\boldsymbol{\theta}'$ satisfies $y_i \Phi(\boldsymbol{\theta}'; \mathbf{x}_i) \geq 1$ for every $i \in \{1, 2\}$, and $\|\boldsymbol{\theta}'\| < \|\tilde{\boldsymbol{\theta}}\|$. Such $\boldsymbol{\theta}'$ is obtained from $\tilde{\boldsymbol{\theta}}$ by slightly changing $\tilde{\mathbf{w}}_1$, $\tilde{\mathbf{w}}_2$, and $\tilde{v}_2$. Thus, by using the second hidden neuron, which is not active in $\tilde{\boldsymbol{\theta}}$, we can obtain a solution $\boldsymbol{\theta}'$ with smaller norm. $\qquad\square$

We note that the assumption on the initialization in the above theorem holds with constant probability for standard initialization schemes (e.g., Xavier initialization).

**Remark 3.1** (Unbounded sub-optimality). *By choosing appropriate inputs $\mathbf{x}_1, \mathbf{x}_2$ in the setting of Thm. 3.2, it is not hard to show that the sub-optimality of the KKT point w.r.t. the global optimum can be arbitrarily large. Namely, for every large $M > 0$ we can choose a dataset where the angle between $\mathbf{x}_1$ and $\mathbf{x}_2$ is sufficiently close to $\pi$, such that $\frac{\|\tilde{\boldsymbol{\theta}}\|}{\|\boldsymbol{\theta}^*\|} \geq M$, where $\tilde{\boldsymbol{\theta}}$ is a KKT point to which GF converges, and $\boldsymbol{\theta}^*$ is a global optimum of Problem 1. Indeed, as illustrated in the figure from the proof idea of Thm. 3.2, if one neuron is active on both inputs and the other neuron is not active on any input, then the active neuron needs to be very large in order to achieve margin $1$, while if each neuron is active on a single input then we can achieve margin $1$ with much smaller parameters. We note that such unbounded sub-optimality can be obtained also in other negative results in this work (in Thm. 4.1, 4.3, 4.4 and 5.4).*

**Remark 3.2** (Robustness to small perturbations). *Thm. 3.2 holds even if we slightly perturb the inputs $\mathbf{x}_1, \mathbf{x}_2$. Thus, it is not sensitive to small changes in the dataset. We note that such robustness to small perturbations can be shown also for the negative results in Thm. 4.1, 4.3, 5.1 and 5.4.*

## 4   DEPTH-2 NETWORKS IN $\mathcal{N}$

In this section we study depth-2 linear and ReLU networks in $\mathcal{N}$. We first show that already for linear networks in $\mathcal{N}_{\text{no-share}}$ (more specifically, for diagonal networks) GF may not converge even to a local optimum.

**Theorem 4.1.** *Let $\Phi$ be a depth-$2$ linear or ReLU diagonal neural network parameterized by $\boldsymbol{\theta} = [\mathbf{w}_1, \mathbf{w}_2]$. Consider minimizing either the exponential or the logistic loss using GF. There exists a dataset $\{(\mathbf{x}, y)\} \subseteq \mathbb{R}^2 \times \{-1, 1\}$ of size $1$ and an initialization $\boldsymbol{\theta}(0)$, such that GF converges to zero loss, and converges in direction to a KKT point $\tilde{\boldsymbol{\theta}}$ of Problem 1 which is not a local optimum.*

*Proof idea (for the complete proof see Appendix C.4).* Let $\mathbf{x} = (1, 2)^\top$ and $y = 1$. Let $\boldsymbol{\theta}(0)$ such that $\mathbf{w}_1(0) = \mathbf{w}_2(0) = (1, 0)^\top$. Recalling that the diagonal network computes the function $\mathbf{x} \mapsto (\mathbf{w}_1 \circ \mathbf{w}_2)^\top \mathbf{x}$ (where $\circ$ is the entry-wise product), we see that the second coordinate remains inactive during training. It is not hard to show that GF converges to the KKT point $\tilde{\boldsymbol{\theta}}$ with $\tilde{\mathbf{w}}_1 = \tilde{\mathbf{w}}_2 = (1, 0)^\top$. However, it is not a local optimum, since for small $\epsilon > 0$ the parameters $\boldsymbol{\theta}' = [\mathbf{w}_1', \mathbf{w}_2']$ with $\mathbf{w}_1' = \mathbf{w}_2' = \left(\sqrt{1-\epsilon}, \sqrt{\frac{\epsilon}{2}}\right)^\top$ satisfy the constraints of Problem 1, and we have $\|\boldsymbol{\theta}'\| < \|\tilde{\boldsymbol{\theta}}\|$. $\qquad\square$

By Thm. 3.2 fully-connected ReLU networks may not converge to a local optimum, and by Thm. 4.1 linear (and ReLU) networks with sparse weights may not converge to a local optimum. In the proofs of both of these negative results, GF converges to a KKT point such that one of the weights vectors of the hidden neurons is zero. However, in practice gradient descent often converges to a network that does not contain such disconnected neurons. Hence, a natural question is whether the negative results hold also in networks that do not contain neurons whose weights vector is zero. In the following theorem we show that in linear networks such an assumption allows us to obtain a positive result. Namely, in depth-2 linear networks in $\mathcal{N}_{\text{no-share}}$, if GF converges in direction to a KKT point of Problem 1 that satisfies this condition, then it is guaranteed to be a global optimum. However, we also show that in ReLU networks assuming that all neurons have non-zero weights is not sufficient.

**Theorem 4.2.** *We have:*

1. *Let $\Phi$ be a depth-2 linear neural network in $\mathcal{N}_{no\text{-}share}$, parameterized by $\boldsymbol{\theta}$. Consider minimizing either the exponential or the logistic loss over a dataset $\{(\mathbf{x}_i, y_i)\}_{i=1}^n$ using GF. Assume that there exists time $t_0$ such that $\mathcal{L}(\boldsymbol{\theta}(t_0)) < 1$, and let $\tilde{\boldsymbol{\theta}}$ be the KKT point of Problem 1 such that $\boldsymbol{\theta}(t)$ converges to $\tilde{\boldsymbol{\theta}}$ in direction (such $\tilde{\boldsymbol{\theta}}$ exists by Thm. 2.1). Assume that in the network*

*parameterized by $\tilde{\theta}$ all hidden neurons have non-zero incoming weights vectors. Then, $\tilde{\theta}$ is a global optimum of Problem 1.*

2. *Let $\Phi$ be a fully-connected depth-2 ReLU network with input dimension 2 and 4 hidden neurons, parameterized by $\theta$. Consider minimizing either the exponential or the logistic loss using GF. There exists a dataset and an initialization $\theta(0)$, such that GF converges to zero loss, and converges in direction to a KKT point $\tilde{\theta}$ of Problem 1, which is not a local optimum, and in the network parameterized by $\tilde{\theta}$ all hidden neurons have non-zero incoming weights.*

*Proof idea (for the complete proof see Appendix C.5).* We give here the proof idea for part (1). Let $k$ be the width of the network. For every $j \in [k]$ we denote by $\mathbf{w}_j$ the incoming weights vector to the $j$-th hidden neuron, and by $v_j$ the outgoing weight. Let $\mathbf{u}_j = v_j \mathbf{w}_j$. We consider an optimization problem over the variables $\mathbf{u}_1, \ldots, \mathbf{u}_k$ where the objective is to minimize $\sum_{j \in [k]} \|\mathbf{u}_j\|$ and the constrains correspond to the constraints of Problem 1. Let $\tilde{\theta} = [\tilde{\mathbf{w}}_1, \ldots, \tilde{\mathbf{w}}_k, \tilde{\mathbf{v}}]$ be the KKT point of Problem 1 to which GF converges in direction. For every $j \in [k]$ we denote $\tilde{\mathbf{u}}_j = \tilde{v}_j \tilde{\mathbf{w}}_j$. We show that $\tilde{\mathbf{u}}_1, \ldots, \tilde{\mathbf{u}}_k$ satisfy the KKT conditions of the aforementioned problem. Since the objective there is convex and the constrains are affine, then it is a global optimum. Finally, we show that it implies global optimality of $\tilde{\theta}$. $\qquad\square$

**Remark 4.1** (Implications on margin maximization in the predictor space for diagonal linear networks). *Thm. 4.1 and 4.2 imply analogous results on diagonal linear networks also in the predictor space. As we discussed in Section 2, Gunasekar et al. (2018b) showed that in depth-2 diagonal linear networks, minimizing $\|\theta\|_2$ under the constraints in Problem 1 implies minimizing $\|\beta\|_1$, where $\beta$ is the corresponding linear predictor. Thm. 4.1 can be easily extended to the predictor space, namely, GF on depth-2 linear diagonal networks might converge to a KKT point $\tilde{\theta}$ of Problem 1, such that the corresponding linear predictor $\tilde{\beta}$ is not a local optimum of the following problem:*

$$\arg\min_{\beta} \|\beta\|_1 \quad s.t. \quad \forall i \in [n] \;\; y_i \langle \beta, \mathbf{x}_i \rangle \geq 1 . \tag{2}$$

*Moreover, by combining part (1) of Thm. 4.2 with the result from Gunasekar et al. (2018b), we deduce that if GF on a depth-2 diagonal linear network converges to a KKT point $\tilde{\theta}$ of Problem 1 with non-zero weights vectors, then the corresponding linear predictor is a global optimum of Problem 2.*

By part (2) of Thm. 4.2, assuming that GF converges to a network without zero neurons is not sufficient for obtaining a positive result in the case of ReLU networks. Hence, we now consider a stronger assumption, namely, that the KKT point $\tilde{\theta}$ is such that for every $\mathbf{x}_i$ in the dataset the inputs to all hidden neurons in the computation $\Phi(\tilde{\theta}; \mathbf{x}_i)$ are non-zero. In the following theorem we show that in depth-2 ReLU networks, if the KKT point satisfies this condition then it is guaranteed to be a local optimum of Problem 1. However, even under this condition it is not necessarily a global optimum. The proof is given in Appendix C.6 and uses ideas from the previous proofs, with some required modifications.

**Theorem 4.3.** *Let $\Phi$ be a depth-2 ReLU network in $\mathcal{N}_{no\text{-}share}$ parameterized by $\theta$. Consider minimizing either the exponential or the logistic loss over a dataset $\{(\mathbf{x}_i, y_i)\}_{i=1}^n$ using GF. Assume that there exists time $t_0$ such that $\mathcal{L}(\theta(t_0)) < 1$, and let $\tilde{\theta}$ be the KKT point of Problem 1 such that $\theta(t)$ converges to $\tilde{\theta}$ in direction (such $\tilde{\theta}$ exists by Thm. 2.1). Assume that for every $i \in [n]$ the inputs to all hidden neurons in the computation $\Phi(\tilde{\theta}; \mathbf{x}_i)$ are non-zero. Then, $\tilde{\theta}$ is a local optimum of Problem 1. However, it may not be a global optimum, even if the network $\Phi$ is fully connected.*

Note that in all the above theorems we do not allow shared weights. We now consider the case of depth-2 linear or ReLU networks in $\mathcal{N}$, where the first layer is convolutional with disjoint patches (and hence has shared weights), and show that GF does not always converge in direction to a local optimum, even when the inputs to all hidden neurons are non-zero (and hence there are no zero weights vectors).

**Theorem 4.4.** *Let $\Phi$ be a depth-2 linear or ReLU network in $\mathcal{N}$, parameterized by $\theta = [\mathbf{w}, \mathbf{v}]$ for $\mathbf{w}, \mathbf{v} \in \mathbb{R}^2$, such that for $\mathbf{x} \in \mathbb{R}^4$ we have $\Phi(\theta; \mathbf{x}) = \sum_{j=1}^2 v_j \sigma(\mathbf{w}^\top \mathbf{x}^{(j)})$ where $\mathbf{x}^{(1)} = (x_1, x_2)$ and $\mathbf{x}^{(2)} = (x_3, x_4)$. Thus, $\Phi$ is a convolutional network with two disjoint patches. Consider minimizing the exponential or the logistic loss using GF. Then, there exists a dataset $\{(\mathbf{x}, y)\}$ of size 1,*

*and an initialization $\boldsymbol{\theta}(0)$, such that GF converges to zero loss, and converges in direction to a KKT point $\tilde{\boldsymbol{\theta}} = [\tilde{\mathbf{w}}, \tilde{\mathbf{v}}]$ of Problem 1 which is not a local optimum. Moreover, $\langle \tilde{\mathbf{w}}, \mathbf{x}^{(j)} \rangle \neq 0$ for $j \in \{1, 2\}$.*

*Proof idea (for the complete proof see Appendix C.7).* Let $\mathbf{x} = \left(4, \frac{1}{\sqrt{2}}, -4, \frac{1}{\sqrt{2}}\right)^{\top}$ and $y = 1$. Let $\boldsymbol{\theta}(0)$ such that $\mathbf{w}(0) = (0, 1)^{\top}$ and $\mathbf{v}(0) = \left(\frac{1}{\sqrt{2}}, \frac{1}{\sqrt{2}}\right)^{\top}$. Since $\mathbf{x}^{(1)}$ and $\mathbf{x}^{(2)}$ are symmetric w.r.t. $\mathbf{w}(0)$, and $\mathbf{v}(0)$ does not break this symmetry, then $\mathbf{w}$ keeps its direction throughout the training. Thus, GF converges in direction to a KKT point $\tilde{\boldsymbol{\theta}}$ where $\tilde{\mathbf{w}} = (0, 1)^{\top}$ and $\tilde{\mathbf{v}} = \left(\frac{1}{\sqrt{2}}, \frac{1}{\sqrt{2}}\right)^{\top}$. It is not a local optimum, since for every small $\epsilon > 0$ the parameters $\boldsymbol{\theta}' = [\mathbf{w}', \mathbf{v}']$ with $\mathbf{w}' = (\sqrt{\epsilon}, 1 - \epsilon)^{\top}$ and $\mathbf{v}' = \left(\frac{1}{\sqrt{2}} + \frac{\sqrt{\epsilon}}{2}, \frac{1}{\sqrt{2}} - \frac{\sqrt{\epsilon}}{2}\right)^{\top}$ satisfy the constraints of Problem 1, and $\|\boldsymbol{\theta}'\| < \|\tilde{\boldsymbol{\theta}}\|$. □

## 5  DEEP NETWORKS IN $\mathcal{N}$

In this section we study the more general case of depth-$m$ neural networks in $\mathcal{N}$, where $m \geq 2$. First, we show that for networks of depth at least $3$ in $\mathcal{N}_{\text{no-share}}$, GF may not converge to a local optimum of Problem 1, for both linear and ReLU networks, and even where there are no zero weights vectors and the inputs to all hidden neurons are non-zero. We prove this claim for diagonal networks.

**Theorem 5.1.** *Let $m \geq 3$. Let $\Phi$ be a depth-$m$ linear or ReLU diagonal neural network parameterized by $\boldsymbol{\theta}$. Consider minimizing either the exponential or the logistic loss using GF. There exists a dataset $\{(\mathbf{x}, y)\} \subseteq \mathbb{R}^2 \times \{-1, 1\}$ of size $1$ and an initialization $\boldsymbol{\theta}(0)$, such that GF converges to zero loss, and converges in direction to a KKT point $\tilde{\boldsymbol{\theta}}$ of Problem 1 which is not a local optimum. Moreover, all inputs to neurons in the computation $\Phi(\tilde{\boldsymbol{\theta}}; \mathbf{x})$ are non-zero.*

*Proof idea (for the complete proof see Appendix C.8).* Let $\mathbf{x} = (1, 1)^{\top}$ and $y = 1$. Consider the initialization $\boldsymbol{\theta}(0)$ where $\mathbf{w}_j(0) = (1, 1)^{\top}$ for every $j \in [m]$. We show that GF converges in direction to a KKT point $\tilde{\boldsymbol{\theta}} = [\tilde{\mathbf{w}}_1, \ldots, \tilde{\mathbf{w}}_m]$ such that $\tilde{\mathbf{w}}_j = \left(2^{-1/m}, 2^{-1/m}\right)^{\top}$ for all $j \in [m]$. Then, we consider the parameters $\boldsymbol{\theta}' = [\mathbf{w}'_1, \ldots, \mathbf{w}'_m]$ such that for every $j \in [m]$ we have $\mathbf{w}'_j = \left(\left(\frac{1+\epsilon}{2}\right)^{1/m}, \left(\frac{1-\epsilon}{2}\right)^{1/m}\right)^{\top}$, and show that if $\epsilon > 0$ is sufficiently small, then $\boldsymbol{\theta}'$ satisfies the constraints in Problem 1 and we have $\|\boldsymbol{\theta}'\| < \|\tilde{\boldsymbol{\theta}}\|$. □

Note that in the case of linear networks, the above result is in contrast to networks with sparse weights of depth 2 that converge to a global optimum by Thm. 4.2, and to fully-connected networks of any depth that converge to a global optimum by Thm. 3.1. In the case of ReLU networks, the above result is in contrast to the case of depth-2 networks studied in Thm. 4.3, where it is guaranteed to converge to a local optimum.

In light of our negative results, we now consider a weaker notion of margin maximization, namely, maximizing the margin for each layer separately. Let $\Phi$ be a network of depth $m$ in $\mathcal{N}$, parameterized by $\boldsymbol{\theta} = [\mathbf{u}^{(l)}]_{l=1}^m$. The maximum-margin problem for a layer $l_0 \in [m]$ w.r.t. $\boldsymbol{\theta}_0 = [\mathbf{u}_0^{(l)}]_{l=1}^m$ is:

$$\min_{\mathbf{u}^{(l_0)}} \frac{1}{2} \left\| \mathbf{u}^{(l_0)} \right\|^2 \quad \text{s.t.} \quad \forall i \in [n] \ \ y_i \Phi(\boldsymbol{\theta}'; \mathbf{x}_i) \geq 1 \ , \tag{3}$$

where $\boldsymbol{\theta}' = [\mathbf{u}_0^{(1)}, \ldots, \mathbf{u}_0^{(l_0-1)}, \mathbf{u}^{(l_0)}, \mathbf{u}_0^{(l_0+1)}, \ldots, \mathbf{u}_0^{(m)}]$. For linear networks we have the following:

**Theorem 5.2.** *Let $m \geq 2$. Let $\Phi$ be any depth-$m$ linear neural network in $\mathcal{N}$, parameterized by $\boldsymbol{\theta} = [\mathbf{u}^{(l)}]_{l=1}^m$. Consider minimizing either the exponential or the logistic loss over a dataset $\{(\mathbf{x}_i, y_i)\}_{i=1}^n$ using GF. Assume that there exists time $t_0$ such that $\mathcal{L}(\boldsymbol{\theta}(t_0)) < 1$. Then, GF converges in direction to a KKT point $\tilde{\boldsymbol{\theta}} = [\tilde{\mathbf{u}}^{(l)}]_{l=1}^m$ of Problem 1, such that for every layer $l \in [m]$ the parameters vector $\tilde{\mathbf{u}}^{(l)}$ is a global optimum of Problem 3 w.r.t. $\tilde{\boldsymbol{\theta}}$.*

The theorem follows by noticing that if $\Phi$ is a linear network, then the constraints in Problem 3 are affine, and its KKT conditions are implied by the KKT conditions of Problem 1. See Appendix C.9 for the formal proof. By Thm. 4.1, 4.4 and 5.1, linear networks in $\mathcal{N}$ might converge to a KKT point

$\tilde{\boldsymbol{\theta}}$ which is not a local optimum of Problem 1. However, by Thm. 5.2 each layer in $\tilde{\boldsymbol{\theta}}$ is a global optimum of Problem 3. Thus, any improvement to $\tilde{\boldsymbol{\theta}}$ requires changing a few layers simultaneously.

While in linear networks GF maximize the margin for each layer separately, in the following theorem we show that this claim does not hold for ReLU networks: Already for fully-connected networks of depth 2 GF may not converge to a local optimum of Problem 3 (see Appendix C.10 for the proof).

**Theorem 5.3.** *Let $\Phi$ be a fully-connected depth-2 ReLU network with input dimension $2$ and $4$ hidden neurons parameterized by $\boldsymbol{\theta}$. Consider minimizing either the exponential or the logistic loss using GF. There exists a dataset and an initialization $\boldsymbol{\theta}(0)$ such that GF converges to zero loss, and converges in direction to a KKT point $\tilde{\boldsymbol{\theta}}$ of Problem 1, such that the weights of the first layer are not a local optimum of Problem 3 w.r.t. $\tilde{\boldsymbol{\theta}}$.*

Finally, we show that in ReLU networks in $\mathcal{N}$ of any depth, if the KKT point to which GF converges in direction is such that the inputs to hidden neurons are non-zero, then it must be a local optimum of Problem 3 (but not necessarily a global optimum). The proof follows the ideas from the proof of Thm. 5.2, with some required modifications, and is given in Appendix C.11.

**Theorem 5.4.** *Let $m \geq 2$. Let $\Phi$ be any depth-$m$ ReLU network in $\mathcal{N}$ parameterized by $\boldsymbol{\theta} = [\mathbf{u}^{(l)}]_{l=1}^m$. Consider minimizing either the exponential or the logistic loss over a dataset $\{(\mathbf{x}_i, y_i)\}_{i=1}^n$ using GF, and assume that there exists time $t_0$ such that $\mathcal{L}(\boldsymbol{\theta}(t_0)) < 1$. Let $\tilde{\boldsymbol{\theta}} = [\tilde{\mathbf{u}}^{(l)}]_{l=1}^m$ be the KKT point of Problem 1 such that $\boldsymbol{\theta}(t)$ converges to $\tilde{\boldsymbol{\theta}}$ in direction (such $\tilde{\boldsymbol{\theta}}$ exists by Thm. 2.1). Let $l \in [m]$ and assume that for every $i \in [n]$ the inputs to all neurons in layers $\geq l$ in the computation $\Phi(\tilde{\boldsymbol{\theta}}; \mathbf{x}_i)$ are non-zero. Then, the parameters vector $\tilde{\mathbf{u}}^{(l)}$ is a local optimum of Problem 3 w.r.t. $\tilde{\boldsymbol{\theta}}$. However, it may not be a global optimum.*

# 6 NON-HOMOGENEOUS NETWORKS

Let $\bar{\gamma}(\boldsymbol{\theta}) := \min_{i \in [n]} y_i \Phi\left(\frac{\boldsymbol{\theta}}{\|\boldsymbol{\theta}\|}; \mathbf{x}_i\right)$ be the *normalized margin*. If $\Phi$ is homogeneous then maximizing $\bar{\gamma}(\boldsymbol{\theta})$ is equivalent to solving Problem 1, i.e., minimizing $\|\boldsymbol{\theta}\|$ under the constraints (cf. Lyu & Li (2019)). In this section we study the normalized margin in non-homogeneous networks.

Lyu & Li (2019) showed under the assumptions from Thm. 2.1, that a smoothed version of the normalized margin is monotonically increasing when training homogeneous networks. More precisely, there is a function $\tilde{\gamma}(\boldsymbol{\theta})$ which is an $O\left(\|\boldsymbol{\theta}\|^{-L}\right)$-additive approximation of $\bar{\gamma}(\boldsymbol{\theta})$, such that $\tilde{\gamma}(\boldsymbol{\theta})$ is monotonically non-decreasing. This result does not apply to non-homogeneous networks, such as networks with skip connections or bias terms. Lyu & Li (2019) observed empirically that the normalized margin is monotonically increasing also when training non-homogeneous networks. However, they did not provide a proof for this phenomenon and left it as an open problem. Their experiments are on training convolutional neural networks (CNN) with bias on MNIST. In the following theorem we show an example for a simple non-homogeneous network where the normalized margin is monotonically *decreasing* during the training. This example implies that in order to obtain such a result for non-homogeneous networks some additional assumptions must be made.

**Theorem 6.1.** *Let $\Phi$ be a depth-2 linear network with input dimension 1, width 1 and a skip connection. Namely, $\Phi$ is parameterized by $\boldsymbol{\theta} = [w, v, u]$ where $w, v, u \in \mathbb{R}$, and $\Phi(\boldsymbol{\theta}; x) = v \cdot w \cdot x + u \cdot x$. Consider the size-1 dataset $\{(1, 1)\}$, and assume that $\boldsymbol{\theta}(0) = [2, 2, 2]$. Then, GF w.r.t. either the exponential loss or the logistic loss converges to zero loss, converges in direction (i.e., $\lim_{t \to \infty} \frac{\boldsymbol{\theta}(t)}{\|\boldsymbol{\theta}(t)\|}$ exists), and the normalized margin is monotonically decreasing during the training, i.e., $\frac{d\bar{\gamma}(\boldsymbol{\theta}(t))}{dt} < 0$ for all $t \geq 0$. Moreover, we have $\bar{\gamma}(\boldsymbol{\theta}(0)) > 0.9$ and $\lim_{t \to \infty} \bar{\gamma}(\boldsymbol{\theta}(t)) = \frac{1}{2}$.*

We note that the proof readily extends in a few directions: It applies also for a depth-2 network without a skip connection but with a bias term in the output neuron. In addition, it also holds for ReLU networks. Finally, the theorem applies also for the smoothed version of the normalized margin considered in Lyu & Li (2019). The proof of the theorem is given in Appendix C.12. Intuitively, note that if $\frac{v}{\|\boldsymbol{\theta}\|}, \frac{w}{\|\boldsymbol{\theta}\|} = 0$ and $\frac{u}{\|\boldsymbol{\theta}\|} = 1$ then $\bar{\gamma}(\boldsymbol{\theta}) = 1$, and if $\frac{u}{\|\boldsymbol{\theta}\|} = 0$ and $\frac{v}{\|\boldsymbol{\theta}\|} = \frac{w}{\|\boldsymbol{\theta}\|} = \frac{1}{\sqrt{2}}$ then $\bar{\gamma}(\boldsymbol{\theta}) = \frac{1}{2}$. Also, since the partial derivative of the loss w.r.t. $v, w$ depends on $w, v$ (respectively) and on $x$, and the partial derivative w.r.t. $u$ depends only on $x$, then $v, w$ grow faster than $u$ during the training. Hence, as $t$ increases, $\frac{u}{\|\boldsymbol{\theta}\|}$ decreases and $\frac{v}{\|\boldsymbol{\theta}\|}, \frac{w}{\|\boldsymbol{\theta}\|}$ increase.

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

## A   MORE RELATED WORK

Soudry et al. (2018) showed that gradient descent on linearly-separable binary classification problems with exponentially-tailed losses (e.g., the exponential loss and the logistic loss), converges to the maximum $\ell_2$-margin direction. This analysis was extended to other loss functions, tighter convergence rates, non-separable data, and variants of gradient-based optimization algorithms (Nacson et al., 2019b; Ji & Telgarsky, 2018b; Ji et al., 2020; Gunasekar et al., 2018a; Shamir, 2020; Ji & Telgarsky, 2021).

As detailed in Section 2, Lyu & Li (2019) and Ji & Telgarsky (2020) showed that GF on homogeneous neural networks with exponential-type losses converge in direction to a KKT point of the maximum margin problem in the parameter space. Similar results under stronger assumptions were previously obtained in Nacson et al. (2019a); Gunasekar et al. (2018b). The implications of margin maximization in the parameter space on the implicit bias in the predictor space for linear neural networks were studied in Gunasekar et al. (2018b) (as detailed in Section 2) and also in Jagadeesan et al. (2021). Margin maximization in the predictor space for fully-connected linear networks was shown by Ji & Telgarsky (2020) (as detailed in Section 2), and similar results under stronger assumptions were previously established in Gunasekar et al. (2018b) and in Ji & Telgarsky (2018a). The implicit bias in the predictor space of diagonal and convolutional linear networks was studied in Gunasekar et al. (2018b); Moroshko et al. (2020); Yun et al. (2020). The implicit bias in infinitely-wide two-layer homogeneous neural networks was studied in Chizat & Bach (2020).

Finally, the implicit bias of neural networks in regression tasks w.r.t. the square loss was also extensively studied in recent years (e.g., Gunasekar et al. (2018c); Razin & Cohen (2020); Arora et al. (2019); Belabbas (2020); Eftekhari & Zygalakis (2020); Li et al. (2018); Ma et al. (2018); Woodworth et al. (2020); Gidel et al. (2019); Li et al. (2020); Yun et al. (2020); Vardi & Shamir (2021); Azulay et al. (2021)). This setting, however, is less relevant for our work.

## B  PRELIMINARIES ON THE KKT CONDITION

Below we review the definition of the KKT condition for non-smooth optimization problems (cf. Lyu & Li (2019); Dutta et al. (2013)).

Let $f : \mathbb{R}^d \to \mathbb{R}$ be a locally Lipschitz function. The Clarke subdifferential (Clarke et al., 2008) at $\mathbf{x} \in \mathbb{R}^d$ is the convex set

$$\partial^\circ f(\mathbf{x}) := \mathrm{conv} \left\{ \lim_{i \to \infty} \nabla f(\mathbf{x}_i) \, \Big| \, \lim_{i \to \infty} \mathbf{x}_i = \mathbf{x}, \ f \text{ is differentiable at } \mathbf{x}_i \right\} .$$

If $f$ is continuously differentiable at $\mathbf{x}$ then $\partial^\circ f(\mathbf{x}) = \{\nabla f(\mathbf{x})\}$.

Consider the following optimization problem

$$\min f(\mathbf{x}) \quad \text{s.t.} \quad \forall n \in [N] \ g_n(\mathbf{x}) \le 0 \,, \tag{4}$$

where $f, g_1, \ldots, g_n : \mathbb{R}^d \to \mathbb{R}$ are locally Lipschitz functions. We say that $\mathbf{x} \in \mathbb{R}^d$ is a feasible point of Problem 4 if $\mathbf{x}$ satisfies $g_n(\mathbf{x}) \le 0$ for all $n \in [N]$. We say that a feasible point $\mathbf{x}$ is a KKT point if there exists $\lambda_1, \ldots, \lambda_N \ge 0$ such that

1. $\mathbf{0} \in \partial^\circ f(\mathbf{x}) + \sum_{n \in [N]} \lambda_n \partial^\circ g_n(\mathbf{x})$;
2. For all $n \in [N]$ we have $\lambda_n g_n(\mathbf{x}) = 0$.

## C  PROOFS

### C.1  AUXILIARY LEMMAS

Throughout our proofs we use the following two lemmas from Du et al. (2018):

**Lemma C.1** (Du et al. (2018)). *Let $m \ge 2$, and let $\Phi$ be a depth-$m$ fully-connected linear or ReLU network parameterized by $\boldsymbol{\theta} = [W_1, \ldots, W_m]$. Suppose that for every $j \in [m]$ we have $W_j \in \mathbb{R}^{d_j \times d_{j-1}}$. Consider minimizing any differentiable loss function (e.g., the exponential or the logistic loss) over a dataset using GF. Then, for every $j \in [m-1]$ at all time $t$ we have*

$$\frac{d}{dt} \left( \|W_j\|_F^2 - \|W_{j+1}\|_F^2 \right) = 0 \,.$$

*Moreover, for every $j \in [m-1]$ and $i \in [d_j]$ we have*

$$\frac{d}{dt} \left( \|W_j[i,:]\|^2 - \|W_{j+1}[:,i]\|^2 \right) = 0 \,,$$

*where $W_j[i,:]$ is the vector of incoming weights to the $i$-th neuron in the $j$-th hidden layer (i.e., the $i$-th row of $W_j$), and $W_{j+1}[:,i]$ is the vector of outgoing weights from this neuron (i.e., the $i$-th column of $W_{j+1}$).*

**Lemma C.2** (Du et al. (2018)). *Let $m \geq 2$, and let $\Phi$ be a depth-$m$ linear or ReLU network in $\mathcal{N}$, parameterized by $\boldsymbol{\theta} = [\mathbf{u}^{(1)}, \ldots, \mathbf{u}^{(m)}]$. Consider minimizing any differentiable loss function (e.g., the exponential or the logistic loss) over a dataset using GF. Then, for every $j \in [m-1]$ at all time $t$ we have*

$$\frac{d}{dt} \left( \left\| \mathbf{u}^{(j)} \right\|^2 - \left\| \mathbf{u}^{(j+1)} \right\|^2 \right) = 0 \, .$$

Note that Lemma C.2 considers a larger family of neural networks since it allows sparse and shared weights, but Lemma C.1 gives a stronger guarantee, since it implies balancedness between the incoming and outgoing weights of each hidden neuron separately. In our proofs we will also need to use a balancedness property for each hidden neuron separately in depth-2 networks with sparse weights. Since this property is not implied by the above lemmas from Du et al. (2018), we now prove it.

Before stating the lemma, let us introduce some required notations. Let $\Phi$ be a depth-2 network in $\mathcal{N}_{\text{no-share}}$. We can always assume w.l.o.g. that the second layer is fully connected, namely, all hidden neurons are connected to the output neuron. Indeed, otherwise we can ignore the neurons that are not connected to the output neuron. For the network $\Phi$ we use the parameterization $\boldsymbol{\theta} = [\mathbf{w}_1, \ldots, \mathbf{w}_k, \mathbf{v}]$, where $k$ is the number of hidden neurons. For every $j \in [k]$ the vector $\mathbf{w}_j \in \mathbb{R}^{p_j}$ is the weights vector of the $j$-th hidden neuron, and we have $1 \leq p_j \leq d$ where $d$ is the input dimention. For an input $\mathbf{x} \in \mathbb{R}^d$ we denote by $\mathbf{x}^j \in \mathbb{R}^{p_j}$ a sub-vector of $\mathbf{x}$, such that $\mathbf{x}^j$ includes the coordinates of $\mathbf{x}$ that are connected to the $j$-th hidden neuron. Thus, given $\mathbf{x}$, the input to the $j$-th hidden neuron is $\langle \mathbf{w}_j, \mathbf{x}^j \rangle$. The vector $\mathbf{v} \in \mathbb{R}^k$ is the weights vector of the second layer. Overall, we have $\Phi(\boldsymbol{\theta}; \mathbf{x}) = \sum_{j \in [k]} v_j \sigma(\mathbf{w}_j^\top \mathbf{x}^j)$.

**Lemma C.3.** *Let $\Phi$ be a depth-2 linear or ReLU network in $\mathcal{N}_{\text{no-share}}$, parameterized by $\boldsymbol{\theta} = [\mathbf{w}_1, \ldots, \mathbf{w}_k, \mathbf{v}]$. Consider minimizing any differentiable loss function (e.g., the exponential or the logistic loss) over a dataset using GF. Then, for every $j \in [k]$ at all time $t$ we have*

$$\frac{d}{dt} \left( \|\mathbf{w}_j\|^2 - v_j^2 \right) = 0 \, .$$

*Proof.* We have

$$\mathcal{L}(\boldsymbol{\theta}) = \sum_{i \in [n]} \ell \left( y_i \Phi(\boldsymbol{\theta}; \mathbf{x}_i) \right) = \sum_{i \in [n]} \ell \left( y_i \sum_{l \in [k]} v_l \sigma(\mathbf{w}_l^\top \mathbf{x}_i^j) \right) \, .$$

Hence

$$\begin{aligned}
\frac{d}{dt} \left( \|\mathbf{w}_j\|^2 \right) &= 2 \langle \mathbf{w}_j, \frac{d\mathbf{w}_j}{dt} \rangle = -2 \langle \mathbf{w}_j, \nabla_{\mathbf{w}_j} \mathcal{L}(\boldsymbol{\theta}) \rangle \\
&= -2 \sum_{i \in [n]} \ell' \left( y_i \sum_{l \in [k]} v_l \sigma(\mathbf{w}_l^\top \mathbf{x}_i^l) \right) \cdot y_i v_j \sigma'(\mathbf{w}_j^\top \mathbf{x}_i^j) \mathbf{w}_j^\top \mathbf{x}_i^j \\
&= -2 \sum_{i \in [n]} \ell' \left( y_i \sum_{l \in [k]} v_l \sigma(\mathbf{w}_l^\top \mathbf{x}_i^l) \right) \cdot y_i v_j \sigma(\mathbf{w}_j^\top \mathbf{x}_i^j) \, .
\end{aligned}$$

Moreover,

$$\begin{aligned}
\frac{d}{dt} \left( v_j^2 \right) &= 2 v_j \frac{dv_j}{dt} = -2 v_j \nabla_{v_j} \mathcal{L}(\boldsymbol{\theta}) \\
&= -2 v_j \sum_{i \in [n]} \ell' \left( y_i \sum_{l \in [k]} v_l \sigma(\mathbf{w}_l^\top \mathbf{x}_i^l) \right) \cdot y_i \sigma(\mathbf{w}_j^\top \mathbf{x}_i^j) \, .
\end{aligned}$$

Hence the lemma follows. $\qquad\square$

Using the above lemma, we show the following:

**Lemma C.4.** *Let $\Phi$ be a depth-2 linear or ReLU network in $\mathcal{N}_{no\text{-}share}$, parameterized by $\boldsymbol{\theta} = [\mathbf{w}_1, \ldots, \mathbf{w}_k, \mathbf{v}]$. Consider minimizing any differentiable loss function (e.g., the exponential or the logistic loss) over a dataset using GF starting from $\boldsymbol{\theta}(0)$. Assume that $\lim_{t\to\infty} \|\boldsymbol{\theta}(t)\| = \infty$ and that $\boldsymbol{\theta}(t)$ converges in direction to $\tilde{\boldsymbol{\theta}} = [\tilde{\mathbf{w}}_1, \ldots, \tilde{\mathbf{w}}_k, \tilde{\mathbf{v}}]$, i.e., $\tilde{\boldsymbol{\theta}} = \|\tilde{\boldsymbol{\theta}}\| \cdot \lim_{t\to\infty} \frac{\boldsymbol{\theta}(t)}{\|\boldsymbol{\theta}(t)\|}$. Then, for every $l \in [k]$ we have $\|\tilde{\mathbf{w}}_l\| = |\tilde{v}_l|$.*

*Proof.* For every $l \in [k]$, let $\Delta_l = \|\mathbf{w}_l(0)\|^2 - v_l(0)^2$. By Lemma C.3, we have for every $l \in [k]$ and $t \geq 0$ that $\|\mathbf{w}_l(t)\|^2 - v_l(t)^2 = \Delta_l$, namely, the differences between the square norms of the incoming and outgoing weights of each hidden neuron remain constant during the training. Hence, we have

$$|\tilde{v}_l| = \|\tilde{\boldsymbol{\theta}}\| \cdot \lim_{t\to\infty} \frac{|v_l(t)|}{\|\boldsymbol{\theta}(t)\|} = \|\tilde{\boldsymbol{\theta}}\| \cdot \lim_{t\to\infty} \frac{\sqrt{\|\mathbf{w}_l(t)\|^2 - \Delta_l}}{\|\boldsymbol{\theta}(t)\|} .$$

Thus, if $\lim_{t\to\infty} \|\mathbf{w}_l(t)\| = \infty$, then we have $|\tilde{v}_l| = \|\tilde{\boldsymbol{\theta}}\| \cdot \lim_{t\to\infty} \frac{\|\mathbf{w}_l(t)\|}{\|\boldsymbol{\theta}(t)\|} = \|\tilde{\mathbf{w}}_l\|$.

Assume now that $\|\mathbf{w}_l(t)\| \not\to \infty$. By the definition of $\tilde{\boldsymbol{\theta}}$ we have $\|\tilde{\mathbf{w}}_l\| = \|\tilde{\boldsymbol{\theta}}\| \cdot \lim_{t\to\infty} \frac{\|\mathbf{w}_l(t)\|}{\|\boldsymbol{\theta}(t)\|}$. Since $\lim_{t\to\infty} \frac{\|\mathbf{w}_l(t)\|}{\|\boldsymbol{\theta}(t)\|}$ exists and $\lim_{t\to\infty} \|\boldsymbol{\theta}(t)\| = \infty$, then we have $\lim_{t\to\infty} \frac{\|\mathbf{w}_l(t)\|}{\|\boldsymbol{\theta}(t)\|} = 0$. Hence, $\lim_{t\to\infty} \frac{|v_l(t)|}{\|\boldsymbol{\theta}(t)\|} = \lim_{t\to\infty} \frac{\sqrt{\|\mathbf{w}_l(t)\|^2 - \Delta_l}}{\|\boldsymbol{\theta}(t)\|} = 0$. Therefore $\|\tilde{\mathbf{w}}_l\| = \tilde{v}_l = 0$. $\square$

## C.2 PROOF OF THM. 3.1

Suppose that the network $\Phi$ is parameterized by $\boldsymbol{\theta} = [W^{(1)}, \ldots, W^{(m)}]$. By Thm. 2.1, GF converges in direction to a KKT point $\tilde{\boldsymbol{\theta}} = [\tilde{W}^{(1)}, \ldots, \tilde{W}^{(m)}]$ of Problem 1. For every $l \in [m]$ let $\Delta_l = \left\|W^{(l)}(0)\right\|_F^2 - \left\|W^{(1)}(0)\right\|_F^2$. By Lemma C.1, we have for every $l \in [m]$ and $t \geq 0$ that

$$\left\|W^{(l)}(t)\right\|_F^2 - \left\|W^{(1)}(t)\right\|_F^2 = \sum_{j=1}^{l-1} \left\|W^{(j+1)}(t)\right\|_F^2 - \left\|W^{(j)}(t)\right\|_F^2 = \sum_{j=1}^{l-1} \left\|W^{(j+1)}(0)\right\|_F^2 - \left\|W^{(j)}(0)\right\|_F^2$$

$$= \left\|W^{(l)}(0)\right\|_F^2 - \left\|W^{(1)}(0)\right\|_F^2 = \Delta_l .$$

Hence, we have

$$\left\|\tilde{W}^{(l)}\right\|_F = \|\tilde{\boldsymbol{\theta}}\| \cdot \lim_{t\to\infty} \frac{\left\|W^{(l)}(t)\right\|_F}{\|\boldsymbol{\theta}(t)\|} = \|\tilde{\boldsymbol{\theta}}\| \cdot \lim_{t\to\infty} \frac{\sqrt{\left\|W^{(1)}(t)\right\|_F^2 + \Delta_l}}{\|\boldsymbol{\theta}(t)\|} .$$

Since by Thm. 2.1 we have $\lim_{t\to\infty} \|\boldsymbol{\theta}(t)\| = \infty$, then $\lim_{t\to\infty} \left\|W^{(1)}(t)\right\|_F = \infty$, and we have

$$\left\|\tilde{W}^{(l)}\right\|_F = \|\tilde{\boldsymbol{\theta}}\| \cdot \lim_{t\to\infty} \frac{\left\|W^{(1)}(t)\right\|_F}{\|\boldsymbol{\theta}(t)\|} = \left\|\tilde{W}^{(1)}\right\|_F := C .$$

By Ji & Telgarsky (2020) (Proposition 4.4), when GF on a fully-connected linear network w.r.t. the exponential loss or the logistic loss converges to zero loss, then we have the following. There are unit vectors $\mathbf{v}_0, \ldots, \mathbf{v}_m$ such that

$$\lim_{t\to\infty} \frac{W^{(l)}(t)}{\left\|W^{(l)}(t)\right\|_F} = \mathbf{v}_l \mathbf{v}_{l-1}^\top$$

for every $l \in [m]$. Moreover, we have $\mathbf{v}_m = 1$, and $\mathbf{v}_0 = \mathbf{u}$ where

$$\mathbf{u} := \arg\max_{\|\mathbf{u}\|=1} \min_{i\in[n]} y_i \mathbf{u}^\top \mathbf{x}_i$$

is the unique linear max margin predictor.

Note that we have

$$\frac{\tilde{W}^{(l)}}{C} = \frac{\tilde{W}^{(l)}}{\left\|\tilde{W}^{(l)}\right\|_F} = \frac{\|\tilde{\boldsymbol{\theta}}\| \cdot \lim_{t\to\infty} \frac{W^{(l)}(t)}{\|\boldsymbol{\theta}(t)\|}}{\|\tilde{\boldsymbol{\theta}}\| \cdot \lim_{t\to\infty} \frac{\left\|W^{(l)}(t)\right\|_F}{\|\boldsymbol{\theta}(t)\|}} = \lim_{t\to\infty} \frac{W^{(l)}(t)}{\left\|W^{(l)}(t)\right\|_F} = \mathbf{v}_l \mathbf{v}_{l-1}^\top .$$

Thus, $\tilde{W}^{(l)} = C\mathbf{v}_l\mathbf{v}_{l-1}^\top$ for every $l \in [m]$.

Let $\tilde{\mathbf{u}} = \tilde{W}^{(m)} \cdot \ldots \cdot \tilde{W}^{(1)} = C^m\mathbf{u}$. Since $\tilde{\boldsymbol{\theta}}$ is a KKT point of Problem 1, we have for every $l \in [m]$

$$\tilde{W}^{(l)} = \sum_{i\in[n]} \lambda_i y_i \frac{\partial\Phi(\tilde{\boldsymbol{\theta}};\mathbf{x}_i)}{\partial W^{(l)}} ,$$

where $\lambda_i \geq 0$ for every $i$, and $\lambda_i = 0$ if $y_i\Phi(\tilde{\boldsymbol{\theta}};\mathbf{x}_i) \neq 1$. Since $\tilde{W}^{(l)}$ are non-zero then there is $i \in [n]$ such that $1 = y_i\Phi(\tilde{\boldsymbol{\theta}};\mathbf{x}_i) = y_i\tilde{\mathbf{u}}^\top\mathbf{x}_i = y_iC^m\mathbf{u}^\top\mathbf{x}_i$. Likewise, since $\tilde{\boldsymbol{\theta}}$ satisfies the constraints of Problem 1, then for every $i \in [n]$ we have $1 \leq y_i\Phi(\tilde{\boldsymbol{\theta}};\mathbf{x}_i) = y_iC^m\mathbf{u}^\top\mathbf{x}_i$. Since, $\mathbf{u}$ is a unit vector that maximized the margin, then we have

$$\|\tilde{\mathbf{u}}\| = C^m = \min\|\mathbf{u}'\| \text{ s.t. } y_i\mathbf{u}'^\top\mathbf{x}_i \geq 1 \text{ for all } i \in [n] . \tag{5}$$

Assume toward contradiction that there is $\boldsymbol{\theta}'$ with $\|\boldsymbol{\theta}'\| < \|\tilde{\boldsymbol{\theta}}\|$ that satisfies the constraints in Problem 1. Let $\mathbf{u}' = W'^{(m)} \cdot \ldots \cdot W'^{(1)}$. By Eq. 5 we have $\|\mathbf{u}'\| \geq \|\tilde{\mathbf{u}}\| = C^m$. Moreover, we have $\|\mathbf{u}'\| = \|W'^{(m)} \cdot \ldots \cdot W'^{(1)}\| \leq \prod_{l\in[m]}\|W'^{(l)}\|_F$ due to the submultiplicativity of the Frobenius norm. Hence $\prod_{l\in[m]}\|W'^{(l)}\|_F \geq C^m$. The following lemma implies that

$$\|\boldsymbol{\theta}'\|^2 = \sum_{l\in[m]}\|W'^{(l)}\|_F^2 \geq m \cdot C^2 = \sum_{l\in[m]}\|\tilde{W}^{(l)}\|_F^2 = \|\tilde{\boldsymbol{\theta}}\|^2$$

in contradiction to our assumption, and thus completes the proof.

**Lemma C.5.** *Let $a_1, \ldots, a_m$ be real numbers such that $\prod_{j\in[m]} a_j \geq C^m$ for some $C \geq 0$. Then $\sum_{j\in[m]} a_j^2 \geq m \cdot C^2$.*

*Proof.* It suffices to prove the claim for the case where $\prod_{j\in[m]} a_j = C^m$. Indeed, if $\prod_{j\in[m]} a_j > C^m$ then we can replace some $a_j$ with an appropriate $a_j'$ such that $|a_j'| < |a_j|$ and we only decrease $\sum_{j\in[m]} a_j^2$. Consider the following problem

$$\min \frac{1}{2}\sum_{j\in[m]} a_j^2 \quad \text{s.t.} \quad \prod_{j\in[m]} a_j = C^m .$$

Using the Lagrange multipliers we obtain that there is some $\lambda \in \mathbb{R}$ such that for every $l \in [m]$ we have $a_l = \lambda \cdot \prod_{j\neq l} a_j$. Thus, $a_l^2 = \lambda \cdot \prod_{j\in[m]} a_j$. It implies that $a_1^2 = \ldots = a_m^2$. Since $\prod_{j\in[m]} a_j = C^m$ then $|a_j| = C$ for every $j \in [m]$. Hence, $\sum_{j\in[m]} a_j^2 = mC^2$. $\square$

### C.3 PROOF OF THM. 3.2

Consider an initialization $\boldsymbol{\theta}(0)$ is such that $\mathbf{w}_1(0)$ satisfies $\langle\mathbf{w}_1(0),\mathbf{x}_1\rangle > 0$ and $\langle\mathbf{w}_1(0),\mathbf{x}_2\rangle > 0$, and $\mathbf{w}_2(0)$ satisfies $\langle\mathbf{w}_2(0),\mathbf{x}_1\rangle < 0$ and $\langle\mathbf{w}_2(0),\mathbf{x}_2\rangle < 0$. Moreover, assume that $v_1(0) > 0$.

Note that for every $\boldsymbol{\theta}$ such that $\langle\mathbf{w}_2,\mathbf{x}_1\rangle < 0$ and $\langle\mathbf{w}_2,\mathbf{x}_2\rangle < 0$ we have

$$\nabla_{\mathbf{w}_2}\mathcal{L}(\boldsymbol{\theta}) = \sum_{i=1}^2 \ell'(y_i\Phi(\boldsymbol{\theta};\mathbf{x}_i)) \cdot y_i\nabla_{\mathbf{w}_2}\Phi(\boldsymbol{\theta};\mathbf{x}_i) = \sum_{i=1}^2 \ell'(y_i\Phi(\boldsymbol{\theta};\mathbf{x}_i)) \cdot y_i\nabla_{\mathbf{w}_2}\left[v_1\sigma(\mathbf{w}_1^\top\mathbf{x}_i) + v_2\sigma(\mathbf{w}_2^\top\mathbf{x}_i)\right]$$

$$= \sum_{i=1}^2 \ell'(y_i\Phi(\boldsymbol{\theta};\mathbf{x}_i)) \cdot y_iv_2\sigma'(\mathbf{w}_2^\top\mathbf{x}_i)\mathbf{x}_i = \mathbf{0} .$$

and

$$\nabla_{v_2}\mathcal{L}(\boldsymbol{\theta}) = \sum_{i=1}^2 \ell'(y_i\Phi(\boldsymbol{\theta};\mathbf{x}_i)) \cdot y_i\nabla_{v_2}\Phi(\boldsymbol{\theta};\mathbf{x}_i) = \sum_{i=1}^2 \ell'(y_i\Phi(\boldsymbol{\theta};\mathbf{x}_i)) \cdot y_i\nabla_{v_2}\left[v_1\sigma(\mathbf{w}_1^\top\mathbf{x}_i) + v_2\sigma(\mathbf{w}_2^\top\mathbf{x}_i)\right]$$

$$= \sum_{i=1}^2 \ell'(y_i\Phi(\boldsymbol{\theta};\mathbf{x}_i)) \cdot y_i\sigma(\mathbf{w}_2^\top\mathbf{x}_i) = 0 .$$

Hence, $\mathbf{w}_2$ and $v_2$ get stuck in their initial values. Moreover, we have

$$\nabla_{v_1}\mathcal{L}(\boldsymbol{\theta}) = \sum_{i=1}^{2} \ell'(y_i\Phi(\boldsymbol{\theta};\mathbf{x}_i)) \cdot y_i\nabla_{v_1}\left[v_1\sigma(\mathbf{w}_1^\top\mathbf{x}_i) + v_2\sigma(\mathbf{w}_2^\top\mathbf{x}_i)\right] = \sum_{i=1}^{2} \ell'(y_i\Phi(\boldsymbol{\theta};\mathbf{x}_i)) \cdot \sigma(\mathbf{w}_1^\top\mathbf{x}_i) \leq 0 \,.$$

Therefore, for every $t \geq 0$ we have $v_1(t) \geq v_1(0) > 0$.

We denote $\mathbf{w}_1 = (w_1[1], w_1[2])$. Since $\langle\mathbf{w}_1(0), \mathbf{x}_j\rangle > 0$ for $j \in \{1, 2\}$ then $w_1[2](0) > 0$. Assume w.l.o.g. that $w_1[1](0) \geq 0$ (the case where $w_1[1](0) \leq 0$ is similar). For every $\mathbf{w}_1$ that satisfies $w_1[2] \geq 0$ and $0 \leq w_1[1] \leq w_1[1](0)$ we have $\langle\mathbf{w}_1, \mathbf{x}_1\rangle > \langle\mathbf{w}_1, \mathbf{x}_2\rangle > 0$. Thus,

$$\begin{aligned}
\nabla_{\mathbf{w}_1}\mathcal{L}(\boldsymbol{\theta}) &= \sum_{i=1}^{2} \ell'(y_i\Phi(\boldsymbol{\theta};\mathbf{x}_i)) \cdot y_i\nabla_{\mathbf{w}_1}\left[v_1\sigma(\mathbf{w}_1^\top\mathbf{x}_i) + v_2\sigma(\mathbf{w}_2^\top\mathbf{x}_i)\right] \\
&= \sum_{i=1}^{2} \ell'(y_i(v_1\sigma(\mathbf{w}_1^\top\mathbf{x}_i) + 0)) \cdot y_iv_1\sigma'(\mathbf{w}_1^\top\mathbf{x}_i)\mathbf{x}_i \\
&= \sum_{i=1}^{2} \ell'(v_1\mathbf{w}_1^\top\mathbf{x}_i) \cdot v_1\mathbf{x}_i \,.
\end{aligned}$$

Since $\ell'$ is negative and monotonically increasing, and since $v_1\mathbf{w}_1^\top\mathbf{x}_1 > v_1\mathbf{w}_1^\top\mathbf{x}_2$, then $\frac{dw_1[1]}{dt} \leq 0$. Also, $\frac{dw_1[2]}{dt} > 0$. Moreover, if $w_1[1] = 0$ then $v_1\mathbf{w}_1^\top\mathbf{x}_1 = v_1\mathbf{w}_1^\top\mathbf{x}_2$ and thus $\frac{dw_1[1]}{dt} = 0$. Hence, for every $t$ we have $w_1[2](t) \geq w_1[2](0) > 0$ and $0 \leq w_1[1](t) \leq w_1[1](0)$.

If $\mathcal{L}(\boldsymbol{\theta}) \geq 1$ then for some $i \in \{1, 2\}$ we have $\ell(y_i\Phi(\boldsymbol{\theta};\mathbf{x}_i)) \geq \frac{1}{2}$ and hence $\ell'(y_i\Phi(\boldsymbol{\theta};\mathbf{x}_i)) \leq c$ for some constant $c < 0$. Since we also have $v_1 \geq v_1(0) > 0$, we have

$$\frac{dw_1[2]}{dt} \geq -c \cdot v_1(0) \cdot \frac{1}{4} \,.$$

Therefore, if the initialization $\boldsymbol{\theta}(0)$ is such that $\mathcal{L}(\boldsymbol{\theta}) \geq 1$ then $w_1[2](t)$ increases at rate at least $\frac{(-c)\cdot v_1(0)}{4}$ while $w_1[1](t)$ remains in $[0, w_1[1](0)]$. Note that for such $w_1[1]$ and $v_1 \geq v_1(0) > 0$, if $w_1[2]$ is sufficiently large then we have $v_1\langle\mathbf{w}_1, \mathbf{x}_i\rangle \geq 1$ for $i \in \{1, 2\}$. Hence, there is some $t_0$ such that $\mathcal{L}(\boldsymbol{\theta}(t_0)) \leq 2\ell(1) < 1$ for both the exponential loss and the logistic loss.

Therefore, by Thm. 2.1 GF converges in direction to a KKT point of Problem 1, and we have $\lim_{t\to\infty}\mathcal{L}(\boldsymbol{\theta}(t)) = 0$ and $\lim_{t\to\infty}\|\boldsymbol{\theta}(t)\| = \infty$. It remains to show that it does not converge in direction to a local optimum of Problem 1.

Let $\bar{\boldsymbol{\theta}} = \lim_{t\to\infty}\frac{\boldsymbol{\theta}(t)}{\|\boldsymbol{\theta}(t)\|}$. We denote $\bar{\boldsymbol{\theta}} = [\bar{\mathbf{w}}_1, \bar{\mathbf{w}}_2, \bar{v}_1, \bar{v}_2]$. We show that $\bar{\mathbf{w}}_1 = \frac{1}{\sqrt{2}}(0, 1)^\top$, $\bar{v}_1 = \frac{1}{\sqrt{2}}$, $\bar{\mathbf{w}}_2 = \mathbf{0}$ and $\bar{v}_2 = 0$. By Lemma C.1, we have for every $t \geq 0$ that $v_1(t)^2 - \|\mathbf{w}_1(t)\|^2 = v_1(0)^2 - \|\mathbf{w}_1(0)\|^2 := \Delta$. Since for every $t$ we have $\mathbf{w}_2(t) = \mathbf{w}_2(0)$ and $v_2(t) = v_2(0)$, and since $\lim_{t\to\infty}\|\boldsymbol{\theta}(t)\| = \infty$ then we have $\lim_{t\to\infty}\|\mathbf{w}_1(t)\| = \infty$ and $\lim_{t\to\infty}|v_1(t)| = \infty$. Also, since $\lim_{t\to\infty}\|\mathbf{w}_1(t)\| = \infty$ and $w_1[1](t) \in [0, w_1[1](0)]$ then $\lim_{t\to\infty}w_1[2](t) = \infty$. Note that

$$\|\boldsymbol{\theta}(t)\| = \sqrt{\|\mathbf{w}_1(t)\|^2 + v_1(t)^2 + \|\mathbf{w}_2(0)\|^2 + v_2(0)^2} = \sqrt{\Delta + 2\|\mathbf{w}_1(t)\|^2 + \|\mathbf{w}_2(0)\|^2 + v_2(0)^2} \,.$$

Since $w_1[1](t) \in [0, w_1[1](0)]$ and $\|\boldsymbol{\theta}(t)\| \to \infty$, we have

$$\bar{\mathbf{w}}_1[1] = \lim_{t\to\infty}\frac{w_1[1](t)}{\|\boldsymbol{\theta}(t)\|} = 0 \,.$$

Moreover,

$$\bar{\mathbf{w}}_1[2] = \lim_{t\to\infty}\frac{w_1[2](t)}{\|\boldsymbol{\theta}(t)\|} = \lim_{t\to\infty}\sqrt{\frac{(w_1[2](t))^2}{\Delta + 2(w_1[1](t))^2 + 2(w_1[2](t))^2 + \|\mathbf{w}_2(0)\|^2 + v_2(0)^2}} = \frac{1}{\sqrt{2}} \,,$$

and

$$\bar{\mathbf{w}}_2 = \lim_{t\to\infty}\frac{\mathbf{w}_2(t)}{\|\boldsymbol{\theta}(t)\|} = \lim_{t\to\infty}\frac{\mathbf{w}_2(0)}{\|\boldsymbol{\theta}(t)\|} = \mathbf{0} \,.$$

Finally, by Lemma C.4 and since $v_1(t) > 0$, we have $\bar{v}_1 = \|\bar{\mathbf{w}}_1\| = \frac{1}{\sqrt{2}}$. By Lemma C.4 we also have $|\bar{v}_2| = \|\bar{\mathbf{w}}_2\| = 0$.

Next, we show that $\bar{\boldsymbol{\theta}}$ does not point at the direction of a local optimum of Problem 1. Let $\tilde{\boldsymbol{\theta}} = [\tilde{\mathbf{w}}_1, \tilde{\mathbf{w}}_2, \tilde{v}_1, \tilde{v}_2]$ be a KKT point of Problem 1 that points at the direction of $\bar{\boldsymbol{\theta}}$. Such $\tilde{\boldsymbol{\theta}}$ exists since $\boldsymbol{\theta}(t)$ converges in direction to a KKT point. Thus, we have $\tilde{\mathbf{w}}_2 = \mathbf{0}$, $\tilde{v}_2 = 0$, $\tilde{\mathbf{w}}_1 = \alpha(0,1)^\top$ and $\tilde{v}_1 = \alpha$ for some $\alpha > 0$. Since $\tilde{\boldsymbol{\theta}}$ satisfies the KKT conditions, we have

$$\tilde{\mathbf{w}}_1 = \sum_{i=1}^{2} \lambda_i \nabla_{\mathbf{w}_1} \left( y_i \Phi(\tilde{\boldsymbol{\theta}}; \mathbf{x}_i) \right) = \sum_{i=1}^{2} \lambda_i y_i \left( \tilde{v}_1 \sigma'(\tilde{\mathbf{w}}_1^\top \mathbf{x}_i) \mathbf{x}_i \right) ,$$

where $\lambda_i \geq 0$ and $\lambda_i = 0$ if $y_i \Phi(\tilde{\boldsymbol{\theta}}; \mathbf{x}_i) \neq 1$. Note that the KKT condition should be w.r.t. the Clarke subdifferential, but since $\tilde{\mathbf{w}}_1^\top \mathbf{x}_i > 0$ for $i \in \{1, 2\}$ then we use here the gradient. Hence, there is $i \in \{1, 2\}$ such that $y_i \Phi(\tilde{\boldsymbol{\theta}}; \mathbf{x}_i) = 1$. Thus,

$$1 = y_i \Phi(\tilde{\boldsymbol{\theta}}; \mathbf{x}_i) = \tilde{v}_1 \sigma(\tilde{\mathbf{w}}_1^\top \mathbf{x}_i) + \tilde{v}_2 \sigma(\tilde{\mathbf{w}}_2^\top \mathbf{x}_i) = \alpha \cdot \frac{\alpha}{4} + 0 = \frac{\alpha^2}{4} .$$

Therefore, $\alpha = 2$ and we have $\tilde{\mathbf{w}}_1 = (0, 2)^\top$ and $\tilde{v}_1 = 2$.

In order to show that $\tilde{\boldsymbol{\theta}}$ is not a local optimum, we show that for every $0 < \epsilon' < 1$ there exists some $\boldsymbol{\theta}'$ such that $\left\| \boldsymbol{\theta}' - \tilde{\boldsymbol{\theta}} \right\| \leq \epsilon'$, $\boldsymbol{\theta}'$ satisfies $\Phi(\boldsymbol{\theta}'; \mathbf{x}_i) \geq 1$ for every $i \in \{1, 2\}$, and $\|\boldsymbol{\theta}'\| < \|\tilde{\boldsymbol{\theta}}\|$. Let $\epsilon = \frac{\epsilon'^2}{9} < \frac{1}{2}$. Let $\boldsymbol{\theta}' = [\mathbf{w}_1', \mathbf{w}_2', v_1', v_2']$ be such that $\mathbf{w}_1' = (\frac{\epsilon}{2}, 2 - 2\epsilon)^\top$, $\mathbf{w}_2' = (-\sqrt{2\epsilon}, 0)^\top$, $v_1' = 2$ and $v_2' = \sqrt{2\epsilon}$. Note that

$$\Phi(\boldsymbol{\theta}'; \mathbf{x}_1) = 2 \cdot \sigma \left( (\frac{\epsilon}{2}, 2 - 2\epsilon)(1, \frac{1}{4})^\top \right) + \sqrt{2\epsilon} \cdot \sigma \left( (-\sqrt{2\epsilon}, 0)(1, \frac{1}{4})^\top \right)$$

$$= 2 \cdot \sigma \left( \frac{\epsilon}{2} + \frac{1}{2} - \frac{\epsilon}{2} \right) + \sqrt{2\epsilon} \cdot \sigma \left( -\sqrt{2\epsilon} \right) = 1 ,$$

and

$$\Phi(\boldsymbol{\theta}'; \mathbf{x}_2) = 2 \cdot \sigma \left( (\frac{\epsilon}{2}, 2 - 2\epsilon)(-1, \frac{1}{4})^\top \right) + \sqrt{2\epsilon} \cdot \sigma \left( (-\sqrt{2\epsilon}, 0)(-1, \frac{1}{4})^\top \right)$$

$$= 2 \cdot \sigma \left( -\frac{\epsilon}{2} + \frac{1}{2} - \frac{\epsilon}{2} \right) + \sqrt{2\epsilon} \cdot \sigma \left( \sqrt{2\epsilon} \right) = 1 - 2\epsilon + 2\epsilon = 1 .$$

We also have

$$\left\| \boldsymbol{\theta}' - \tilde{\boldsymbol{\theta}} \right\|^2 = \|\mathbf{w}_1' - \tilde{\mathbf{w}}_1\|^2 + \|\mathbf{w}_2' - \tilde{\mathbf{w}}_2\|^2 + (v_1' - \tilde{v}_1)^2 + (v_2' - \tilde{v}_2)^2$$

$$= \left( \frac{\epsilon^2}{4} + 4\epsilon^2 \right) + 2\epsilon + 0 + 2\epsilon < 9\epsilon = \epsilon'^2 .$$

Finally, we have

$$\|\boldsymbol{\theta}'\|^2 = \frac{\epsilon^2}{4} + 4 - 8\epsilon + 4\epsilon^2 + 2\epsilon + 4 + 2\epsilon = 8 - 4\epsilon + \frac{17\epsilon^2}{4} < 8 - 4\epsilon + \frac{17\epsilon}{8} < 8 = \left\| \tilde{\boldsymbol{\theta}} \right\|^2 .$$

Thus, $\|\boldsymbol{\theta}'\| < \|\tilde{\boldsymbol{\theta}}\|$.

## C.4 PROOF OF THM. 4.1

Let $\mathbf{x} = (1, 2)^\top$ and $y = 1$. Let $\boldsymbol{\theta}(0)$ such that $\mathbf{w}_1(0) = \mathbf{w}_2(0) = (1, 0)^\top$. Note that $\mathcal{L}(\boldsymbol{\theta}(0)) = \ell(1) < 1$ for both linear and ReLU networks with the exponential loss or the logistic loss, and therefore by Thm. 2.1 GF converges in direction to a KKT point $\tilde{\boldsymbol{\theta}}$ of Problem 1, and we have $\lim_{t \to \infty} \mathcal{L}(\boldsymbol{\theta}(t)) = 0$ and $\lim_{t \to \infty} \|\boldsymbol{\theta}(t)\| = \infty$. We denote $\mathbf{w}_1 = (\mathbf{w}_1[1], \mathbf{w}_1[2])^\top$ and $\mathbf{w}_2 = (\mathbf{w}_2[1], \mathbf{w}_2[2])^\top$. Note that the initialization $\boldsymbol{\theta}(0)$ is such that the second hidden neuron has 0 in both its incoming and outgoing weights. Hence, the gradient w.r.t. $\mathbf{w}_1[2]$ and $\mathbf{w}_2[2]$ is zero, and the second hidden neuron remains inactive during the training. Moreover, $\mathbf{w}_1[1]$ and $\mathbf{w}_2[1]$ are strictly

increasing. Also, by Lemma C.3 we have for every $t \geq 0$ that $\mathbf{w}_1[1](t)^2 = \mathbf{w}_2[1](t)^2$. Overall, $\tilde{\boldsymbol{\theta}}$ is such that $\tilde{\mathbf{w}}_1 = \tilde{\mathbf{w}}_2 = (1,0)^\top$. Note that since the dataset is of size 1, then every KKT point of Problem 1 must label the input $\mathbf{x}$ with exactly 1.

It remains to show that $\tilde{\boldsymbol{\theta}}$ is not local optimum. Let $0 < \epsilon < 1$, and let $\boldsymbol{\theta}' = [\mathbf{w}'_1, \mathbf{w}'_2]$ with $\mathbf{w}'_1 = \mathbf{w}'_2 = \left(\sqrt{1-\epsilon}, \sqrt{\frac{\epsilon}{2}}\right)^\top$. Note that $\boldsymbol{\theta}'$ satisfies the constraints of Problem 1, since $y \cdot \Phi(\boldsymbol{\theta}'; \mathbf{x}) = 1 - \epsilon + 2 \cdot \frac{\epsilon}{2} = 1$. Moreover, we have $\|\tilde{\boldsymbol{\theta}}\|^2 = 2$ and $\|\boldsymbol{\theta}'\|^2 = 2\left(1 - \epsilon + \frac{\epsilon}{2}\right) = 2 - \epsilon$ and therefore $\|\boldsymbol{\theta}'\| < \|\tilde{\boldsymbol{\theta}}\|$.

## C.5 Proof of Thm. 4.2

### C.5.1 Proof of part 1

We assume w.l.o.g. that the second layer is fully-connected, namely, all hidden neurons are connected to the output neuron, since otherwise we can ignore disconnected neurons. For the network $\Phi$ we use the parameterization $\boldsymbol{\theta} = [\mathbf{w}_1, \ldots, \mathbf{w}_k, \mathbf{v}]$ introduced in Section C.1. Thus, we have $\Phi(\boldsymbol{\theta}; \mathbf{x}) = \sum_{l \in [k]} v_l \mathbf{w}_l^\top \mathbf{x}^l$.

By Thm. 2.1, GF converges in direction to $\tilde{\boldsymbol{\theta}} = [\tilde{\mathbf{w}}_1, \ldots, \tilde{\mathbf{w}}_k, \tilde{\mathbf{v}}]$ which satisfies the KKT conditions of Problem 1. Thus, there are $\lambda_1, \ldots, \lambda_n$ such that for every $j \in [k]$ we have

$$\tilde{\mathbf{w}}_j = \sum_{i \in [n]} \lambda_i \nabla_{\mathbf{w}_j} \left(y_i \Phi(\tilde{\boldsymbol{\theta}}; \mathbf{x}_i)\right) = \sum_{i \in [n]} \lambda_i y_i \tilde{v}_j \mathbf{x}_i^j , \tag{6}$$

and we have $\lambda_i \geq 0$ for all $i$, and $\lambda_i = 0$ if $y_i \Phi(\tilde{\boldsymbol{\theta}}; \mathbf{x}_i) = y_i \sum_{l \in [k]} \tilde{v}_l \tilde{\mathbf{w}}_l^\top \mathbf{x}_i^l \neq 1$. By Thm. 2.1, we also have $\lim_{t \to \infty} \|\boldsymbol{\theta}(t)\| = \infty$. Hence, by Lemma C.4 we have $\|\tilde{\mathbf{w}}_j\| = |\tilde{v}_j|$ for all $j \in [k]$.

Consider the following problem

$$\min \sum_{l \in [k]} \|\mathbf{u}_l\| \quad \text{s.t.} \quad \forall i \in [n] \ \ y_i \sum_{l \in [k]} \mathbf{u}_l^\top \mathbf{x}_i^l \geq 1 . \tag{7}$$

For every $l \in [k]$ we denote $\tilde{\mathbf{u}}_l = \tilde{v}_l \cdot \tilde{\mathbf{w}}_l$. Since we assume that $\tilde{\mathbf{w}}_l \neq \mathbf{0}$ for every $l \in [k]$, and since $\|\tilde{\mathbf{w}}_l\| = |\tilde{v}_l|$, then $\tilde{\mathbf{u}}_l \neq \mathbf{0}$ for all $l \in [k]$. Note that since $\tilde{\mathbf{w}}_1, \ldots, \tilde{\mathbf{w}}_k, \tilde{\mathbf{v}}$ satisfy the constraints in Problem 1, then $\tilde{\mathbf{u}}_1, \ldots, \tilde{\mathbf{u}}_k$ satisfy the constraints in the above problem. In order to show that $\tilde{\mathbf{u}}_1, \ldots, \tilde{\mathbf{u}}_k$ satisfy the KKT condition of the problem, we need to prove that for every $j \in [k]$ we have

$$\frac{\tilde{\mathbf{u}}_j}{\|\tilde{\mathbf{u}}_j\|} = \sum_{i \in [n]} \lambda'_i y_i \mathbf{x}_i^j \tag{8}$$

for some $\lambda'_i \geq 0$ such that $\lambda'_i = 0$ if $y_i \sum_{l \in [k]} \tilde{\mathbf{u}}_l^\top \mathbf{x}_i^l \neq 1$. From Eq. 6 and since $\|\tilde{\mathbf{w}}_l\| = |\tilde{v}_l|$ for every $l \in [k]$, we have

$$\tilde{\mathbf{u}}_j = \tilde{v}_j \cdot \tilde{\mathbf{w}}_j = \tilde{v}_j \sum_{i \in [n]} \lambda_i y_i \tilde{v}_j \mathbf{x}_i^j = \tilde{v}_j^2 \sum_{i \in [n]} \lambda_i y_i \mathbf{x}_i^j = \|\tilde{v}_j \tilde{\mathbf{w}}_j\| \sum_{i \in [n]} \lambda_i y_i \mathbf{x}_i^j = \|\tilde{\mathbf{u}}_j\| \sum_{i \in [n]} \lambda_i y_i \mathbf{x}_i^j .$$

Note that we have $\lambda_i \geq 0$ for all $i$, and $\lambda_i = 0$ if $y_i \sum_{l \in [k]} \tilde{\mathbf{u}}_l^\top \mathbf{x}_i^l = y_i \sum_{l \in [k]} \tilde{v}_l \tilde{\mathbf{w}}_l^\top \mathbf{x}_i^l \neq 1$. Hence Eq. 8 holds with $\lambda'_1, \ldots, \lambda'_n$ that satisfy the requirement. Since the objective in Problem 7 is convex and the constraints are affine functions, then its KKT condition is sufficient for global optimality. Namely, $\tilde{\mathbf{u}}_1, \ldots, \tilde{\mathbf{u}}_k$ are a global optimum for problem 7.

We now deduce that $\tilde{\boldsymbol{\theta}}$ is a global optimum for Problem 1. Assume toward contradiction that there is a solution $\boldsymbol{\theta}' = [\mathbf{w}'_1, \ldots, \mathbf{w}'_k, \mathbf{v}']$ for the constraints in Problem 1 such that $\|\boldsymbol{\theta}'\|^2 < \|\tilde{\boldsymbol{\theta}}\|^2$. Let $\mathbf{u}'_l = v'_l \mathbf{w}'_l$. Note that the vectors $\mathbf{u}'_l$ satisfy the constraints in Problem 7. Moreover, we have

$$\sum_{l \in [k]} \|\mathbf{u}'_l\| = \sum_{l \in [k]} |v'_l| \cdot \|\mathbf{w}'_l\| \leq \sum_{l \in [k]} \frac{1}{2}\left(|v'_l|^2 + \|\mathbf{w}'_l\|^2\right) = \frac{1}{2}\|\boldsymbol{\theta}'\|^2 < \frac{1}{2}\left\|\tilde{\boldsymbol{\theta}}\right\|^2 = \sum_{l \in [k]} \frac{1}{2}\left(|\tilde{v}_l|^2 + \|\tilde{\mathbf{w}}_l\|^2\right) .$$

Since $\|\tilde{\mathbf{w}}_l\| = |\tilde{v}_l|$, the above equals

$$\sum_{l \in [k]} \|\tilde{\mathbf{w}}_l\|^2 = \sum_{l \in [k]} |\tilde{v}_l| \cdot \|\tilde{\mathbf{w}}_l\| = \sum_{l \in [k]} \|\tilde{\mathbf{u}}_l\| ,$$

which contradicts the global optimality of $\tilde{\mathbf{u}}_1, \ldots, \tilde{\mathbf{u}}_k$.

### C.5.2 PROOF OF PART 2

Let $\{(\mathbf{x}_i, y_i)\}_{i=1}^4$ be a dataset such that $y_i = 1$ for all $i \in [4]$ and we have $\mathbf{x}_1 = (0,1)^\top$, $\mathbf{x}_2 = (1,0)^\top$, $\mathbf{x}_3 = (0,-1)$ and $\mathbf{x}_4 = (-1,0)$. Consider the initialization $\boldsymbol{\theta}(0) = [\mathbf{w}_1(0), \mathbf{w}_2(0), \mathbf{w}_3(0), \mathbf{w}_4(0), \mathbf{v}(0)]$ such that $\mathbf{w}_i(0) = 2\mathbf{x}_i$ and $v_i(0) = 2$ for every $i \in [4]$. Note that $\mathcal{L}(\boldsymbol{\theta}(0)) = 4\ell(4) < 1$ for both the exponential loss and the logistic loss, and therefore by Thm. 2.1 GF converges in direction to a KKT point $\tilde{\boldsymbol{\theta}}$ of Problem 1, and we have $\lim_{t\to\infty} \mathcal{L}(\boldsymbol{\theta}(t)) = 0$ and $\lim_{t\to\infty} \|\boldsymbol{\theta}(t)\| = \infty$.

We now show that for all $t \geq 0$ we have $\mathbf{w}_i(t) = \alpha(t)\mathbf{x}_i$ and $v_i(t) = \alpha(t)$ where $\alpha(t) > 0$ and $\lim_{t\to\infty} \alpha(t) = \infty$. Indeed, for such $\boldsymbol{\theta}(t)$, for every $j \in [4]$ we have

$$-\frac{d\mathbf{w}_j}{dt} = \nabla_{\mathbf{w}_j}\mathcal{L}(\boldsymbol{\theta}) = \sum_{i=1}^4 \ell'(y_i\Phi(\boldsymbol{\theta};\mathbf{x}_i)) \cdot y_i \nabla_{\mathbf{w}_j}\Phi(\boldsymbol{\theta};\mathbf{x}_i) = \sum_{i=1}^4 \ell'\left(\sum_{l=1}^4 v_l\sigma(\mathbf{w}_l^\top\mathbf{x}_i)\right) \cdot \left(v_j\sigma'(\mathbf{w}_j^\top\mathbf{x}_i)\mathbf{x}_i\right)$$

$$= \ell'(\alpha^2) \cdot \alpha \cdot \sum_{i=1}^4 \sigma'(\mathbf{w}_j^\top\mathbf{x}_i)\mathbf{x}_i = \ell'(\alpha^2) \cdot \alpha\mathbf{x}_j ,$$

and

$$-\frac{dv_j}{dt} = \nabla_{v_j}\mathcal{L}(\boldsymbol{\theta}) = \sum_{i=1}^4 \ell'(y_i\Phi(\boldsymbol{\theta};\mathbf{x}_i)) \cdot y_i \nabla_{v_j}\Phi(\boldsymbol{\theta};\mathbf{x}_i) = \sum_{i=1}^4 \ell'\left(\sum_{l=1}^4 v_l\sigma(\mathbf{w}_l^\top\mathbf{x}_i)\right) \cdot \sigma(\mathbf{w}_j^\top\mathbf{x}_i)$$

$$= \ell'(\alpha^2) \cdot \sum_{i=1}^4 \sigma(\mathbf{w}_j^\top\mathbf{x}_i) = \ell'(\alpha^2) \cdot \alpha .$$

Moreover, since $\lim_{t\to\infty} \|\boldsymbol{\theta}(t)\| = \infty$ then $\lim_{t\to\infty} \alpha(t) = \infty$.

Hence, the KKT point $\tilde{\boldsymbol{\theta}}$ is such that for every $j \in [4]$ the vector $\tilde{\mathbf{w}}_j$ points at the direction $\mathbf{x}_j$, and we have $\tilde{v}_j = \|\tilde{\mathbf{w}}_j\|$. Also, the vectors $\tilde{\mathbf{w}}_1, \tilde{\mathbf{w}}_2, \tilde{\mathbf{w}}_3, \tilde{\mathbf{w}}_4$ have equal norms. That is, $\tilde{\mathbf{w}}_j = \tilde{\alpha}\mathbf{x}_j$ and $\tilde{v}_j = \tilde{\alpha}$ for some $\tilde{\alpha} > 0$. Moreover, since it satisfies the KKT condition of Problem 1, then we have

$$\tilde{\mathbf{w}}_j = \sum_{i=1}^4 \lambda_i y_i \nabla_{\mathbf{w}_j}\Phi(\tilde{\boldsymbol{\theta}};\mathbf{x}_i) ,$$

where $\lambda_i \geq 0$ and $\lambda_i = 0$ if $y_i\Phi(\tilde{\boldsymbol{\theta}};\mathbf{x}_i) \neq 1$. Hence, there is $i$ such that $y_i\Phi(\tilde{\boldsymbol{\theta}};\mathbf{x}_i) = 1$. Therefore, $\tilde{\alpha}^2 = 1$. Thus, we conclude that for all $j \in [4]$ we have $\tilde{\mathbf{w}}_j = \mathbf{x}_j$ and $\tilde{v}_j = 1$. Note that $\tilde{\mathbf{w}}_j \neq \mathbf{0}$ for all $j \in [4]$ as required.

Next, we show that $\tilde{\boldsymbol{\theta}}$ is not a local optimum of Problem 1. We show that for every $0 < \epsilon < 1$ there exists some $\boldsymbol{\theta}'$ such that $\left\|\boldsymbol{\theta}' - \tilde{\boldsymbol{\theta}}\right\| \leq \epsilon$, $\boldsymbol{\theta}'$ satisfies the constraints of Problem 1, and $\|\boldsymbol{\theta}'\| < \|\tilde{\boldsymbol{\theta}}\|$. Let $\epsilon' = \frac{\epsilon}{2\sqrt{2}}$. Let $\boldsymbol{\theta}'$ be such that $v_j' = \tilde{v}_j = 1$ for all $j \in [4]$, and we have $\mathbf{w}_1' = (\epsilon', 1-\epsilon')^\top$, $\mathbf{w}_2' = (1-\epsilon', -\epsilon')^\top$, $\mathbf{w}_3' = (-\epsilon', -1+\epsilon')^\top$ and $\mathbf{w}_4' = (-1+\epsilon', \epsilon')^\top$. It is easy to verify that $\boldsymbol{\theta}'$ satisfies the constraints. Indeed, we have $\Phi(\boldsymbol{\theta}';\mathbf{x}_i) = ((1-\epsilon') + \epsilon' + 0 + 0) = 1$. Also, we have $\left\|\boldsymbol{\theta}' - \tilde{\boldsymbol{\theta}}\right\| = \sqrt{4 \cdot 2\epsilon'^2} = 2\sqrt{2}\epsilon' = \epsilon$. Finally,

$$\|\boldsymbol{\theta}'\|^2 = 4 \cdot \left(\epsilon'^2 + (1-\epsilon')^2\right) + 4 = 8 + 8\epsilon'\left(\epsilon'-1\right) < 8 = \|\tilde{\boldsymbol{\theta}}\|^2 .$$

### C.6 PROOF OF THM. 4.3

We assume w.l.o.g. that the second layer is fully-connected, namely, all hidden neurons are connected to the output neuron, since otherwise we can ignore disconnected neurons. For the network $\Phi$ we use the parameterization $\boldsymbol{\theta} = [\mathbf{w}_1, \ldots, \mathbf{w}_k, \mathbf{v}]$ introduced in Section C.1. Thus, we have $\Phi(\boldsymbol{\theta};\mathbf{x}) = \sum_{l\in[k]} v_l\sigma(\mathbf{w}_l^\top\mathbf{x}^l)$.

We denote $\tilde{\boldsymbol{\theta}} = [\tilde{\mathbf{w}}_1, \ldots, \tilde{\mathbf{w}}_k, \tilde{\mathbf{v}}]$. Since $\tilde{\boldsymbol{\theta}}$ is a KKT point of Problem 1, then there are $\lambda_1, \ldots, \lambda_n$ such that for every $j \in [k]$ we have

$$\tilde{\mathbf{w}}_j = \sum_{i\in[n]} \lambda_i \nabla_{\mathbf{w}_j}\left(y_i\Phi(\tilde{\boldsymbol{\theta}};\mathbf{x}_i)\right) = \sum_{i\in[n]} \lambda_i y_i \tilde{v}_j \sigma'(\tilde{\mathbf{w}}_j^\top\mathbf{x}_i^j)\mathbf{x}_i^j , \tag{9}$$

and we have $\lambda_i \geq 0$ for all $i$, and $\lambda_i = 0$ if $y_i \Phi(\tilde{\boldsymbol{\theta}}; \mathbf{x}_i) = y_i \sum_{l \in [k]} \tilde{v}_l \sigma(\tilde{\mathbf{w}}_l^\top \mathbf{x}_i) \neq 1$. Note that the KKT condition should be w.r.t. the Clarke subdifferential, but since for all $i, j$ we have $\tilde{\mathbf{w}}_j^\top \mathbf{x}_i^j \neq 0$ by our assumption, then we can use here the gradient. By Thm. 2.1, we also have $\lim_{t \to \infty} \|\boldsymbol{\theta}(t)\| = \infty$. Hence, by Lemma C.4 we have $\|\tilde{\mathbf{w}}_j\| = |\tilde{v}_j|$ for all $j \in [k]$.

For $i \in [n]$ and $j \in [k]$ let $A_{ij} = \mathbb{1}(\tilde{\mathbf{w}}_j^\top \mathbf{x}_i^j \geq 0)$. Consider the following problem

$$\min \sum_{l \in [k]} \|\mathbf{u}_l\| \quad \text{s.t.} \quad \forall i \in [n] \ \ y_i \sum_{l \in [k]} A_{il} \mathbf{u}_l^\top \mathbf{x}_i^l \geq 1 . \tag{10}$$

For every $l \in [k]$ let $\tilde{\mathbf{u}}_l = \tilde{v}_l \cdot \tilde{\mathbf{w}}_l$. Since we assume that the inputs to all neurons in the computations $\Phi(\tilde{\boldsymbol{\theta}}; \mathbf{x}_i)$ are non-zero, then we must have $\tilde{\mathbf{w}}_l \neq \mathbf{0}$ for every $l \in [k]$. Since we also have $\|\tilde{\mathbf{w}}_l\| = |\tilde{v}_l|$, then $\tilde{\mathbf{u}}_l \neq \mathbf{0}$ for all $l \in [k]$. Note that since $\tilde{\mathbf{w}}_1, \ldots, \tilde{\mathbf{w}}_k, \tilde{\mathbf{v}}$ satisfy the constraints in Problem 1, then $\tilde{\mathbf{u}}_1, \ldots, \tilde{\mathbf{u}}_k$ satisfy the constraints in the above problem. Indeed, for every $i \in [n]$ we have

$$y_i \sum_{l \in [k]} A_{il} \tilde{\mathbf{u}}_l^\top \mathbf{x}_i^l = y_i \sum_{l \in [k]} \mathbb{1}(\tilde{\mathbf{w}}_l^\top \mathbf{x}_i^l \geq 0) \tilde{v}_l \tilde{\mathbf{w}}_l^\top \mathbf{x}_i^l = y_i \sum_{l \in [k]} \tilde{v}_l \sigma(\tilde{\mathbf{w}}_l^\top \mathbf{x}_i^l) \geq 1 .$$

In order to show that $\tilde{\mathbf{u}}_1, \ldots, \tilde{\mathbf{u}}_k$ satisfy the KKT condition of Problem 10, we need to prove that for every $j \in [k]$ we have

$$\frac{\tilde{\mathbf{u}}_j}{\|\tilde{\mathbf{u}}_j\|} = \sum_{i \in [n]} \lambda_i' y_i A_{ij} \mathbf{x}_i^j \tag{11}$$

for some $\lambda_1', \ldots, \lambda_n'$ such that for all $i$ we have $\lambda_i' \geq 0$, and $\lambda_i' = 0$ if $y_i \sum_{l \in [k]} A_{il} \tilde{\mathbf{u}}_l^\top \mathbf{x}_i^l \neq 1$. From Eq. 9 and since $\|\tilde{\mathbf{w}}_l\| = |\tilde{v}_l|$ for every $l \in [k]$, we have

$$\tilde{\mathbf{u}}_j = \tilde{v}_j \cdot \tilde{\mathbf{w}}_j = \tilde{v}_j \sum_{i \in [n]} \lambda_i y_i \tilde{v}_j A_{ij} \mathbf{x}_i^j = \tilde{v}_j^2 \sum_{i \in [n]} \lambda_i y_i A_{ij} \mathbf{x}_i^j = \|\tilde{v}_j \tilde{\mathbf{w}}_j\| \sum_{i \in [n]} \lambda_i y_i A_{ij} \mathbf{x}_i^j = \|\tilde{\mathbf{u}}_j\| \sum_{i \in [n]} \lambda_i y_i A_{ij} \mathbf{x}_i^j .$$

Note that we have $\lambda_i \geq 0$ for all $i$, and $\lambda_i = 0$ if

$$y_i \sum_{l \in [k]} A_{il} \tilde{\mathbf{u}}_l^\top \mathbf{x}_i^l = y_i \sum_{l \in [k]} \tilde{v}_l \mathbb{1}(\tilde{\mathbf{w}}_l^\top \mathbf{x}_i^l \geq 0) \tilde{\mathbf{w}}_l^\top \mathbf{x}_i^l = y_i \sum_{l \in [k]} \tilde{v}_l \sigma(\tilde{\mathbf{w}}_l^\top \mathbf{x}_i^l) \neq 1 .$$

Hence Eq. 11 holds with $\lambda_1', \ldots, \lambda_n'$ that satisfy the requirement. Since the objective in Problem 10 is convex and the constraints are affine functions, then its KKT condition is sufficient for global optimality. Namely, $\tilde{\mathbf{u}}_1, \ldots, \tilde{\mathbf{u}}_k$ are a global optimum for Problem 10.

We now deduce that $\tilde{\boldsymbol{\theta}}$ is a local optimum for Problem 1. Since for every $i \in [n]$ and $l \in [k]$ we have $\tilde{\mathbf{w}}_l^\top \mathbf{x}_i^l \neq 0$, then there is $\epsilon > 0$, such that for every $i, l$ and every $\mathbf{w}_l'$ with $\|\mathbf{w}_l' - \tilde{\mathbf{w}}_l\| \leq \epsilon$ we have $\mathbb{1}(\tilde{\mathbf{w}}_l^\top \mathbf{x}_i^l \geq 0) = \mathbb{1}(\mathbf{w}_l'^\top \mathbf{x}_i^l \geq 0)$. Assume toward contradiction that there is a solution $\boldsymbol{\theta}' = [\mathbf{w}_1', \ldots, \mathbf{w}_k', \mathbf{v}']$ for the constraints in Problem 1 such that $\|\boldsymbol{\theta}' - \tilde{\boldsymbol{\theta}}\| \leq \epsilon$ and $\|\boldsymbol{\theta}'\|^2 < \|\tilde{\boldsymbol{\theta}}\|^2$. Note that we have $\|\mathbf{w}_l' - \tilde{\mathbf{w}}_l\| \leq \epsilon$ for every $l \in [k]$. We denote $\mathbf{u}_l' = v_l' \mathbf{w}_l'$. The vectors $\mathbf{u}_1', \ldots, \mathbf{u}_k'$ satisfy the constraints in Problem 10, since we have

$$y_i \sum_{l \in [k]} A_{il} \mathbf{u}_l'^\top \mathbf{x}_i^l = y_i \sum_{l \in [k]} \mathbb{1}(\tilde{\mathbf{w}}_l^\top \mathbf{x}_i^l \geq 0) v_l' \mathbf{w}_l'^\top \mathbf{x}_i^l = y_i \sum_{l \in [k]} \mathbb{1}(\mathbf{w}_l'^\top \mathbf{x}_i^l \geq 0) v_l' \mathbf{w}_l'^\top \mathbf{x}_i^l$$

$$= y_i \sum_{l \in [k]} v_l' \sigma(\mathbf{w}_l'^\top \mathbf{x}_i^l) \geq 1 ,$$

where the last inequality is since $\boldsymbol{\theta}'$ satisfies the constraints in Problem 1. Moreover, we have

$$\sum_{l \in [k]} \|\mathbf{u}_l'\| = \sum_{l \in [k]} |v_l'| \cdot \|\mathbf{w}_l'\| \leq \sum_{l \in [k]} \frac{1}{2} \left( |v_l'|^2 + \|\mathbf{w}_l'\|^2 \right) = \frac{1}{2} \|\boldsymbol{\theta}'\|^2 < \frac{1}{2} \left\| \tilde{\boldsymbol{\theta}} \right\|^2 = \sum_{l \in [k]} \frac{1}{2} \left( |\tilde{v}_l|^2 + \|\tilde{\mathbf{w}}_l\|^2 \right) .$$

Since $\|\tilde{\mathbf{w}}_l\| = |\tilde{v}_l|$, the above equals

$$\sum_{l \in [k]} \|\tilde{\mathbf{w}}_l\|^2 = \sum_{l \in [k]} |\tilde{v}_l| \cdot \|\tilde{\mathbf{w}}_l\| = \sum_{l \in [k]} \|\tilde{\mathbf{u}}_l\| ,$$

which contradicts the global optimality of $\tilde{\mathbf{u}}_1, \ldots, \tilde{\mathbf{u}}_k$.

It remains to show that $\tilde{\boldsymbol{\theta}}$ may not be a global optimum of Problem 1, even if the network $\Phi$ is fully connected. The following lemma concludes the proof.

**Lemma C.6.** *Let $\Phi$ be a depth-2 fully-connected ReLU network with input dimension $2$ and two hidden neurons. Consider minimizing either the exponential or the logistic loss using GF. Then, there exists a dataset $\{(\mathbf{x}_i, y_i)\}_{i=1}^n$ and an initialization $\boldsymbol{\theta}(0)$, such that GF converges to zero loss, converges in direction to a KKT point $\tilde{\boldsymbol{\theta}} = [\tilde{\mathbf{w}}_1, \tilde{\mathbf{w}}_2, \tilde{\mathbf{v}}]$ of Problem 1 such that $\langle \tilde{\mathbf{w}}_j, \mathbf{x}_i \rangle \neq 0$ for all $j \in \{1, 2\}$ and $i \in [n]$, and $\tilde{\boldsymbol{\theta}}$ is not a global optimum.*

*Proof.* Let $\mathbf{x}_1 = \left(1, \frac{1}{4}\right)^\top$, $\mathbf{x}_2 = \left(-1, \frac{1}{4}\right)^\top$, $\mathbf{x}_3 = (0, -1)$, $y_1 = y_2 = y_3 = 1$. Let $\{(\mathbf{x}_1, y_1), (\mathbf{x}_2, y_2), (\mathbf{x}_3, y_3)\}$ be a dataset. Consider the initialization $\boldsymbol{\theta}(0)$ such that $\mathbf{w}_1(0) = (0, 3)$, $v_1(0) = 3$, $\mathbf{w}_2(0) = (0, -2)$ and $v_2(0) = 2$. Note that $\mathcal{L}(\boldsymbol{\theta}(0)) = 2\ell\left(\frac{9}{4}\right) + \ell(4) < 1$ for both the exponential loss and the logistic loss, and therefore by Thm. 2.1 GF converges in direction to a KKT point $\tilde{\boldsymbol{\theta}}$ of Problem 1.

Note that for $\boldsymbol{\theta}$ such that $\mathbf{w}_1 = \alpha \cdot (0, 1)^\top$ and $\mathbf{w}_2 = \beta \cdot (0, -1)^\top$ for some $\alpha, \beta > 0$, and $v_1, v_2 > 0$, we have

$$\nabla_{\mathbf{w}_1}\mathcal{L}(\boldsymbol{\theta}) = \sum_{i=1}^3 \ell'(y_i\Phi(\boldsymbol{\theta}; \mathbf{x}_i)) \cdot y_i \nabla_{\mathbf{w}_1}\Phi(\boldsymbol{\theta}; \mathbf{x}_i) = \sum_{i=1}^3 \ell'\left(v_1\sigma(\mathbf{w}_1^\top \mathbf{x}_i) + v_2\sigma(\mathbf{w}_2^\top \mathbf{x}_i)\right) \cdot v_1\sigma'(\mathbf{w}_1^\top \mathbf{x}_i)\mathbf{x}_i$$
$$= \sum_{i=1}^2 \ell'(v_1\sigma(\mathbf{w}_1^\top \mathbf{x}_i)) \cdot v_1\mathbf{x}_i = v_1\ell'\left(v_1\frac{\alpha}{4}\right)\sum_{i=1}^2 \mathbf{x}_i \ ,$$

and

$$\nabla_{v_1}\mathcal{L}(\boldsymbol{\theta}) = \sum_{i=1}^3 \ell'(y_i\Phi(\boldsymbol{\theta}; \mathbf{x}_i)) \cdot y_i \nabla_{v_1}\Phi(\boldsymbol{\theta}; \mathbf{x}_i) = \sum_{i=1}^3 \ell'(y_i\Phi(\boldsymbol{\theta}; \mathbf{x}_i)) \cdot \sigma(\mathbf{w}_1^\top \mathbf{x}_i) \ .$$

Hence, $-\nabla_{\mathbf{w}_1}\mathcal{L}(\boldsymbol{\theta})$ points in the direction $(0, 1)^\top$ and $-\nabla_{v_1}\mathcal{L}(\boldsymbol{\theta}) > 0$. Moreover, we have

$$\nabla_{\mathbf{w}_2}\mathcal{L}(\boldsymbol{\theta}) = \sum_{i=1}^3 \ell'(y_i\Phi(\boldsymbol{\theta}; \mathbf{x}_i)) \cdot y_i \nabla_{\mathbf{w}_2}\Phi(\boldsymbol{\theta}; \mathbf{x}_i) = \sum_{i=1}^3 \ell'(v_1\sigma(\mathbf{w}_1^\top \mathbf{x}_i) + v_2\sigma(\mathbf{w}_2^\top \mathbf{x}_i)) \cdot v_2\sigma'(\mathbf{w}_2^\top \mathbf{x}_i)\mathbf{x}_i$$
$$= \ell'(v_2\sigma(\mathbf{w}_2^\top \mathbf{x}_3)) \cdot v_2\mathbf{x}_3 = v_2\ell'(v_2\beta)\mathbf{x}_3 \ ,$$

and

$$\nabla_{v_2}\mathcal{L}(\boldsymbol{\theta}) = \sum_{i=1}^3 \ell'(y_i\Phi(\boldsymbol{\theta}; \mathbf{x}_i)) \cdot y_i \nabla_{v_2}\Phi(\boldsymbol{\theta}; \mathbf{x}_i) = \sum_{i=1}^3 \ell'(v_1\sigma(\mathbf{w}_1^\top \mathbf{x}_i) + v_2\sigma(\mathbf{w}_2^\top \mathbf{x}_i)) \cdot \sigma(\mathbf{w}_2^\top \mathbf{x}_i)$$
$$= \ell'(v_2\sigma(\mathbf{w}_2^\top \mathbf{x}_3))\sigma(\mathbf{w}_2^\top \mathbf{x}_3) = \ell'(v_2\beta) \cdot \beta \ .$$

Therefore, $-\nabla_{\mathbf{w}_2}\mathcal{L}(\boldsymbol{\theta})$ points in the direction $(0, -1)^\top$ and $-\nabla_{v_2}\mathcal{L}(\boldsymbol{\theta}) > 0$. Hence for every $t$ we have $\mathbf{w}_1(t) = \alpha(t) \cdot (0, 1)^\top$ for some $\alpha(t) > 0$ and $v_1(t) > 0$. Also, we have $\mathbf{w}_2(t) = \beta(t) \cdot (0, -1)^\top$ for some $\beta(t) > 0$ and $v_2(t) > 0$. By Lemma C.1, we have for every $t \geq 0$ that $\|\mathbf{w}_1(t)\|^2 - v_1(t)^2 = \|\mathbf{w}_1(0)\|^2 - v_1(0)^2 = 0$ and $\|\mathbf{w}_2(t)\|^2 - v_2(t)^2 = \|\mathbf{w}_2(0)\|^2 - v_2(0)^2 = 0$. Hence, we have $v_1(t) = \alpha(t)$ and $v_2(t) = \beta(t)$. Therefore, we have $\tilde{\mathbf{w}}_1 = \tilde{\alpha} \cdot (0, 1)^\top$ and $\tilde{v}_1 = \tilde{\alpha}$ for some $\tilde{\alpha} \geq 0$. Likewise, we have $\tilde{\mathbf{w}}_2 = \tilde{\beta} \cdot (0, -1)^\top$ and $\tilde{v}_2 = \tilde{\beta}$ for some $\tilde{\beta} \geq 0$. Since $\tilde{\boldsymbol{\theta}}$ satisfies the constraints in Problem 1, then $\tilde{\alpha} \geq 2$ and $\tilde{\beta} \geq 1$. Note that $\langle \tilde{\mathbf{w}}_j, \mathbf{x}_i \rangle \neq 0$ for all $j \in \{1, 2\}$ and $i \in \{1, 2, 3\}$.

We now show that there exists a solution $\boldsymbol{\theta}'$ to Problem 1 with a smaller norm, and hence $\tilde{\boldsymbol{\theta}}$ is not a global optimum. Let $\boldsymbol{\theta}' = [\mathbf{w}_1', \mathbf{w}_2', \mathbf{v}']$ such that $\mathbf{w}_1' = \frac{\mathbf{x}_1}{\tilde{\alpha}\|\mathbf{x}_1\|}$, $v_1' = \tilde{\alpha}$, $\mathbf{w}_2' = \frac{1}{\tilde{\beta}} \cdot \left(-\frac{5}{4}, -1\right)$, and $v_2' = \tilde{\beta}$. It is easy to verify that $\boldsymbol{\theta}'$ satisfies the constraints in Problem 1, and we have

$$\|\boldsymbol{\theta}'\|^2 = \frac{1}{\tilde{\alpha}^2} + \tilde{\alpha}^2 + \frac{1}{\tilde{\beta}^2}\left(\frac{25}{16} + 1\right) + \tilde{\beta}^2 < \frac{1}{4} + \tilde{\alpha}^2 + 1 \cdot 3 + \tilde{\beta}^2 < \tilde{\beta}^2 + \tilde{\alpha}^2 + \tilde{\alpha}^2 + \tilde{\beta}^2 = \|\tilde{\boldsymbol{\theta}}\|^2 \ .$$

$\square$

## C.7 Proof of Thm. 4.4

Let $\mathbf{x} = \left(4, \frac{1}{\sqrt{2}}, -4, \frac{1}{\sqrt{2}}\right)^\top$ and $y = 1$. Let $\boldsymbol{\theta}(0) = [\mathbf{w}(0), \mathbf{v}(0)]$ where $\mathbf{w}(0) = (0, 1)^\top$ and $\mathbf{v}(0) = \left(\frac{1}{\sqrt{2}}, \frac{1}{\sqrt{2}}\right)^\top$. Note that $\Psi(\boldsymbol{\theta}(0); \mathbf{x}) = 1$ and hence $\mathcal{L}(\boldsymbol{\theta}(0)) < 1$ for both the exponential loss and the logistic loss. Therefore, by Thm. 2.1 GF converges in direction to a KKT point $\tilde{\boldsymbol{\theta}}$ of Problem 1, and we have $\lim_{t \to \infty} \mathcal{L}(\boldsymbol{\theta}(t)) = 0$ and $\lim_{t \to \infty} \|\boldsymbol{\theta}(t)\| = \infty$.

The symmetry of the input $\mathbf{x}$ and the initialization $\boldsymbol{\theta}(0)$ implies that the direction of $\mathbf{w}$ does not change during the training, and that we have $v_1(t) = v_2(t) > 0$ for all $t \geq 0$. More formally, this claim follows from the following calculation. For $j \in \{1, 2\}$ we have

$$\nabla_{v_j} \mathcal{L}(\boldsymbol{\theta}) = \ell'(y\Phi(\boldsymbol{\theta}; \mathbf{x})) \cdot y \nabla_{v_j} \Phi(\boldsymbol{\theta}; \mathbf{x}) = \ell'(y\Phi(\boldsymbol{\theta}; \mathbf{x})) \cdot \sigma(\mathbf{w}^\top \mathbf{x}^{(j)}) .$$

Moreover,

$$\nabla_{\mathbf{w}} \mathcal{L}(\boldsymbol{\theta}) = \ell'(y\Phi(\boldsymbol{\theta}; \mathbf{x})) \cdot y \nabla_{\mathbf{w}} \Phi(\boldsymbol{\theta}; \mathbf{x}) = \ell'(y\Phi(\boldsymbol{\theta}; \mathbf{x})) \cdot \left(v_1 \sigma'(\mathbf{w}^\top \mathbf{x}^{(1)}) \mathbf{x}^{(1)} + v_2 \sigma'(\mathbf{w}^\top \mathbf{x}^{(2)}) \mathbf{x}^{(2)}\right) .$$

Hence, if $v_1 = v_2 > 0$ and $\mathbf{w}$ points in the direction $(0, 1)^\top$, then it is easy to verify that $\nabla_{v_1} \mathcal{L}(\boldsymbol{\theta}) = \nabla_{v_1} \mathcal{L}(\boldsymbol{\theta}) < 0$ and that $\nabla_{\mathbf{w}} \mathcal{L}(\boldsymbol{\theta})$ points in the direction of $-(\mathbf{x}^{(1)} + \mathbf{x}^{(2)}) = -(0, \sqrt{2})^\top$. Furthermore, by Lemma C.2, for every $t \geq 0$ we have $\|\mathbf{w}(t)\|^2 - \|\mathbf{v}(t)\|^2 = \|\mathbf{w}(0)\|^2 - \|\mathbf{v}(0)\|^2 = 0$.

Therefore, the KKT point $\tilde{\boldsymbol{\theta}} = [\tilde{\mathbf{w}}, \tilde{\mathbf{v}}]$ is such that $\tilde{\mathbf{w}}$ points at the direction $(0, 1)^\top$, $\tilde{v}_1 = \tilde{v}_2 > 0$, and $\|\tilde{\mathbf{w}}\| = \|\tilde{\mathbf{v}}\|$. Since $\tilde{\boldsymbol{\theta}}$ satisfies the KKT conditions of Problem 1, then we have

$$\tilde{\mathbf{w}} = \lambda \nabla_{\mathbf{w}} \left(y\Phi(\tilde{\boldsymbol{\theta}}; \mathbf{x})\right) ,$$

where $\lambda \geq 0$ and $\lambda = 0$ if $y\Phi(\tilde{\boldsymbol{\theta}}; \mathbf{x}) \neq 1$. Hence, we must have $y\Phi(\tilde{\boldsymbol{\theta}}; \mathbf{x}) = 1$. Letting $z := \tilde{v}_1 = \tilde{v}_2$ and using $2z^2 = \|\tilde{\mathbf{v}}\|^2 = \|\tilde{\mathbf{w}}\|^2 = \tilde{w}_2^2$, we have

$$1 = \tilde{v}_1 \sigma(\tilde{\mathbf{w}}^\top \mathbf{x}^{(1)}) + \tilde{v}_2 \sigma(\tilde{\mathbf{w}}^\top \mathbf{x}^{(2)}) = z\tilde{\mathbf{w}}^\top \mathbf{x}^{(1)} + z\tilde{\mathbf{w}}^\top \mathbf{x}^{(2)} = z\tilde{\mathbf{w}}^\top \left(\mathbf{x}^{(1)} + \mathbf{x}^{(2)}\right) = z \cdot \tilde{w}_2 \sqrt{2}$$

$$= \frac{\tilde{w}_2}{\sqrt{2}} \cdot \tilde{w}_2 \sqrt{2} = \tilde{w}_2^2 .$$

Therefore, $\tilde{\mathbf{w}} = (0, 1)^\top$ and $\tilde{\mathbf{v}} = \left(\frac{1}{\sqrt{2}}, \frac{1}{\sqrt{2}}\right)$. Note that we have $\langle \tilde{\mathbf{w}}, \mathbf{x}^{(1)} \rangle \neq 0$ and $\langle \tilde{\mathbf{w}}, \mathbf{x}^{(2)} \rangle \neq 0$.

It remains to show that $\tilde{\boldsymbol{\theta}}$ is not a local optimum of Problem 1. We show that for every $0 < \epsilon' < 1$ there exists some $\boldsymbol{\theta}' = [\mathbf{w}', \mathbf{v}']$ such that $\left\|\boldsymbol{\theta}' - \tilde{\boldsymbol{\theta}}\right\| \leq \epsilon'$, $\boldsymbol{\theta}'$ satisfies the constrains in Problem 1, and $\|\boldsymbol{\theta}'\| < \|\tilde{\boldsymbol{\theta}}\|$. Let $\epsilon = \frac{\epsilon'^2}{2} \in (0, 1/2)$, and let $\mathbf{w}' = (\sqrt{\epsilon}, 1 - \epsilon)^\top$ and $\mathbf{v}' = \left(\frac{1}{\sqrt{2}} + \frac{\sqrt{\epsilon}}{2}, \frac{1}{\sqrt{2}} - \frac{\sqrt{\epsilon}}{2}\right)^\top$. Note that

$$\|\boldsymbol{\theta}'\|^2 = \|\mathbf{w}'\|^2 + \|\mathbf{v}'\|^2 = \epsilon + (1 - \epsilon)^2 + \left(\frac{1}{\sqrt{2}} + \frac{\sqrt{\epsilon}}{2}\right)^2 + \left(\frac{1}{\sqrt{2}} - \frac{\sqrt{\epsilon}}{2}\right)^2 = \epsilon + 1 + \epsilon^2 - 2\epsilon + 1 + \frac{\epsilon}{2}$$

$$= 2 - \frac{\epsilon}{2} + \epsilon^2 < 2 - \frac{\epsilon}{2} + \frac{\epsilon}{2} = 2 = \|\tilde{\mathbf{w}}\|^2 + \|\tilde{\mathbf{v}}\|^2 = \left\|\tilde{\boldsymbol{\theta}}\right\|^2 .$$

Moreover,

$$\left\|\boldsymbol{\theta}' - \tilde{\boldsymbol{\theta}}\right\|^2 = \epsilon + \epsilon^2 + \frac{\epsilon}{4} + \frac{\epsilon}{4} = \epsilon^2 + \frac{3\epsilon}{2} = \frac{\epsilon'^4}{4} + \frac{3\epsilon'^2}{4} < \frac{\epsilon'^2}{4} + \frac{3\epsilon'^2}{4} = \epsilon'^2 .$$

Finally, we show that $\boldsymbol{\theta}'$ satisfies the constraints:

$$\Phi(\boldsymbol{\theta}'; \mathbf{x}) = v_1' \sigma(\mathbf{w}'^\top \mathbf{x}^{(1)}) + v_2' \sigma(\mathbf{w}'^\top \mathbf{x}^{(2)})$$

$$= \left(\frac{1}{\sqrt{2}} + \frac{\sqrt{\epsilon}}{2}\right) \left(4\sqrt{\epsilon} + \frac{1}{\sqrt{2}} \cdot (1 - \epsilon)\right) + \left(\frac{1}{\sqrt{2}} - \frac{\sqrt{\epsilon}}{2}\right) \left(-4\sqrt{\epsilon} + \frac{1}{\sqrt{2}} \cdot (1 - \epsilon)\right)$$

$$= \frac{1}{\sqrt{2}} \cdot (1 - \epsilon) \left(\frac{1}{\sqrt{2}} + \frac{\sqrt{\epsilon}}{2} + \frac{1}{\sqrt{2}} - \frac{\sqrt{\epsilon}}{2}\right) + 4\sqrt{\epsilon} \left(\frac{1}{\sqrt{2}} + \frac{\sqrt{\epsilon}}{2} - \frac{1}{\sqrt{2}} + \frac{\sqrt{\epsilon}}{2}\right)$$

$$= 1 - \epsilon + 4\epsilon = 1 + 3\epsilon \geq 1 .$$

## C.8 PROOF OF THM. 5.1

Let $\mathbf{x} = (1,1)^\top$ and $y = 1$. Consider the initialization $\boldsymbol{\theta}(0) = [\mathbf{w}_1(0), \ldots, \mathbf{w}_m(0)]$, where $\mathbf{w}_j(0) = (1,1)^\top$ for every $j \in [m]$. Note that $\mathcal{L}(\boldsymbol{\theta}(0)) = \ell(2) < 1$ for both linear and ReLU networks with the exponential loss or the logistic loss, and therefore by Thm. 2.1 GF converges in direction to a KKT point $\tilde{\boldsymbol{\theta}}$ of Problem 1, and we have $\lim_{t\to\infty} \mathcal{L}(\boldsymbol{\theta}(t)) = 0$ and $\lim_{t\to\infty} \|\boldsymbol{\theta}(t)\| = \infty$. It remains to show that it does not converge in direction to a local optimum of Problem 1.

From the symmetry of the network $\Phi$ and the initialization $\boldsymbol{\theta}(0)$, it follows that for all $t$ the network $\Phi(\boldsymbol{\theta}(t); \cdot)$ remains symmetric, namely, there are $\alpha_j(t)$ such that $\mathbf{w}_j(t) = (\alpha_j(t), \alpha_j(t))$. Moreover, by Lemma C.2, for every $t \geq 0$ and $j, l \in [m]$ we have $\alpha_j(t) = \alpha_l(t) := \alpha(t)$. Thus, GF converges in direction to the KKT point $\tilde{\boldsymbol{\theta}} = [\tilde{\mathbf{w}}_1, \ldots, \tilde{\mathbf{w}}_m]$ such that $\tilde{\mathbf{w}}_j = \left(2^{-1/m}, 2^{-1/m}\right)^\top$ for all $j \in [m]$. Note that since the dataset is of size 1, then every KKT point of Problem 1 must label the input $\mathbf{x}$ with exactly 1.

We now show that $\tilde{\boldsymbol{\theta}}$ is not a local optimum of Problem 1. The following arguments hold for both linear and ReLU networks. Let $0 < \epsilon < \frac{1}{2}$. Let $\boldsymbol{\theta}' = [\mathbf{w}_1', \ldots, \mathbf{w}_m']$ such that for every $j \in [m]$ we have $\mathbf{w}_j' = \left(\left(\frac{1+\epsilon}{2}\right)^{1/m}, \left(\frac{1-\epsilon}{2}\right)^{1/m}\right)^\top$. We have

$$y \cdot \Phi(\boldsymbol{\theta}'; \mathbf{x}) = \left(\frac{1+\epsilon}{2}\right) + \left(\frac{1-\epsilon}{2}\right) = 1 .$$

Hence, $\boldsymbol{\theta}'$ satisfies the constraints in Problem 1. We now show that for every sufficiently small $\epsilon > 0$ we have $\|\boldsymbol{\theta}'\|^2 < \|\tilde{\boldsymbol{\theta}}\|^2$. We need to show that

$$m\left(\frac{1+\epsilon}{2}\right)^{2/m} + m\left(\frac{1-\epsilon}{2}\right)^{2/m} < 2m\left(\frac{1}{2}\right)^{2/m} .$$

Therefore, it suffices to show that

$$(1+\epsilon)^{2/m} + (1-\epsilon)^{2/m} < 2 .$$

Let $g : \mathbb{R} \to \mathbb{R}$ such that $g(s) = (1+s)^{2/m} + (1-s)^{2/m}$. We have $g(0) = 2$. The derivatives of $g$ satisfy

$$g'(s) = \frac{2}{m}(1+s)^{\frac{2}{m}-1} - \frac{2}{m}(1-s)^{\frac{2}{m}-1} ,$$

and

$$g''(s) = \frac{2}{m}\left(\frac{2}{m}-1\right)(1+s)^{\frac{2}{m}-2} + \frac{2}{m}\left(\frac{2}{m}-1\right)(1-s)^{\frac{2}{m}-2} .$$

Since $m \geq 3$ we have $g'(0) = 0$ and $g''(0) < 0$. Hence, $0$ is a local maximum of $g$. Therefore for every sufficiently small $\epsilon > 0$ we have $g(\epsilon) < 2$ and thus $\|\boldsymbol{\theta}'\|^2 < \|\tilde{\boldsymbol{\theta}}\|^2$.

Finally, note that the inputs to all neurons in the computation $\Phi(\tilde{\boldsymbol{\theta}}; \mathbf{x})$ are positive.

## C.9 PROOF OF THM. 5.2

By Thm. 2.1 GF converge in direction to a KKT point $\tilde{\boldsymbol{\theta}} = [\tilde{\mathbf{u}}^{(l)}]_{l=1}^m$ of Problem 1. We now show that for every layer $l \in [m]$ the parameters vector $\tilde{\mathbf{u}}^{(l)}$ is a global optimum of Problem 3 w.r.t. $\tilde{\boldsymbol{\theta}}$.

Since $\tilde{\boldsymbol{\theta}}$ is a KKT point of Problem 1, then there are $\lambda_1, \ldots, \lambda_n$ such that for every $l \in [m]$ we have

$$\tilde{\mathbf{u}}^{(l)} = \sum_{i\in[n]} \lambda_i \frac{\partial\left(y_i\Phi(\tilde{\boldsymbol{\theta}}; \mathbf{x}_i)\right)}{\partial\mathbf{u}^{(l)}} ,$$

where $\lambda_i \geq 0$ for all $i$, and $\lambda_i = 0$ if $y_i\Phi(\tilde{\boldsymbol{\theta}}; \mathbf{x}_i) \neq 1$. Letting $\boldsymbol{\theta}'(\mathbf{u}^{(l)}) = [\tilde{\mathbf{u}}^{(1)}, \ldots, \tilde{\mathbf{u}}^{(l-1)}, \mathbf{u}^{(l)}, \tilde{\mathbf{u}}^{(l+1)}, \ldots, \tilde{\mathbf{u}}^{(m)}]$, the above equation can be written as

$$\tilde{\mathbf{u}}^{(l)} = \sum_{i\in[n]} \lambda_i \frac{\partial\left(y_i\Phi(\boldsymbol{\theta}'(\tilde{\mathbf{u}}^{(l)}); \mathbf{x}_i)\right)}{\partial\mathbf{u}^{(l)}} ,$$

where $\lambda_i \geq 0$ for all $i$, and $\lambda_i = 0$ if $y_i \Phi(\boldsymbol{\theta}'(\tilde{\mathbf{u}}^{(l)}); \mathbf{x}_i) = y_i \Phi(\tilde{\boldsymbol{\theta}}; \mathbf{x}_i) \neq 1$. Moreover, if the constraints in Problem 1 are satisfies in $\tilde{\boldsymbol{\theta}}$, then the constrains in Problem 3 are also satisfied for every $l \in [m]$ in $\tilde{\mathbf{u}}^{(l)}$ w.r.t. $\tilde{\boldsymbol{\theta}}$. Hence, for every $l \in [m]$ the KKT conditions of Problem 3 w.r.t. $\tilde{\boldsymbol{\theta}}$ hold. Since the constraints in Problem 3 are affine and the objective is convex, then this KKT point is a global optimum.

## C.10  PROOF OF THM. 5.3

Let $\{(\mathbf{x}_i, y_i)\}_{i=1}^4$ be a dataset such that $y_i = 1$ for all $i \in [4]$ and we have $\mathbf{x}_1 = (0, 1)^\top$, $\mathbf{x}_2 = (1, 0)^\top$, $\mathbf{x}_3 = (0, -1)$ and $\mathbf{x}_4 = (-1, 0)$. In the proof of Thm. 4.2 (part 2) we showed that for an appropriate initialization, for both the exponential loss and the logistic loss GF converges to zero loss, and converges in direction to a KKT point $\tilde{\boldsymbol{\theta}}$ of Problem 1. Moreover, in the proof of Thm. 4.2 we showed that the KKT point $\tilde{\boldsymbol{\theta}}$ is such that for all $j \in [4]$ we have $\tilde{\mathbf{w}}_j = \mathbf{x}_j$ and $\tilde{v}_j = 1$.

We show that $\tilde{\mathbf{w}}_1, \tilde{\mathbf{w}}_2, \tilde{\mathbf{w}}_3, \tilde{\mathbf{w}}_4$ is not a local optimum of Problem 3 w.r.t. $\tilde{\boldsymbol{\theta}}$. It suffices to prove that for every $0 < \epsilon < 1$ there exists some $\boldsymbol{\theta}'$ such that $v'_j = \tilde{v}_j$ for all $j \in [4]$, $\|\boldsymbol{\theta}' - \tilde{\boldsymbol{\theta}}\| \leq \epsilon$, $\boldsymbol{\theta}'$ satisfies the constraints, and $\|\boldsymbol{\theta}'\| < \|\tilde{\boldsymbol{\theta}}\|$. The existence of such $\boldsymbol{\theta}'$ is shown in the proof of Thm. 4.2. Hence, we conclude the proof of the theorem.

## C.11  PROOF OF THM. 5.4

By Thm. 2.1 GF converge in direction to a KKT point $\tilde{\boldsymbol{\theta}} = [\tilde{\mathbf{u}}^{(l)}]_{l=1}^m$ of Problem 1. Let $l \in [m]$ and assume that for every $i \in [n]$ the inputs to all neurons in layers $l, \ldots, m - 1$ in the computation $\Phi(\tilde{\boldsymbol{\theta}}; \mathbf{x}_i)$ are non-zero. We now show that the parameters vector $\tilde{\mathbf{u}}^{(l)}$ is a local optimum of Problem 3 w.r.t. $\tilde{\boldsymbol{\theta}}$.

For $i \in [n]$ and $k \in [m-1]$ we denote by $\mathbf{x}_i^{(k)} \in \mathbb{R}^{d_k}$ the output of the $k$-th layer in the computation $\Phi(\tilde{\boldsymbol{\theta}}; \mathbf{x}_i)$, and denote $\mathbf{x}_i^{(0)} = \mathbf{x}_i$. If $l \in [m-1]$ then we define the following notations. We denote by $f_l : \mathbb{R}^{d_l} \to \mathbb{R}$ the function computed by layers $l + 1, \ldots, m$ of $\Phi(\tilde{\boldsymbol{\theta}}; \cdot)$. Thus, we have $\Phi(\tilde{\boldsymbol{\theta}}; \mathbf{x}_i) = f_l(\mathbf{x}_i^{(l)}) = f_l \circ \sigma\left(\tilde{W}^{(l)} \mathbf{x}_i^{(l-1)}\right)$, where $\tilde{W}^{(l)}$ is the weight matrix that corresponds to $\tilde{\mathbf{u}}^{(l)}$. For $i \in [n]$ we denote by $h_i$ the function $\mathbf{u}^{(l)} \mapsto f_l \circ \sigma(W^{(l)} \mathbf{x}_i^{(l-1)})$ where $W^{(l)}$ is the weights matrix that corresponds to $\mathbf{u}^{(l)}$. Thus, $\Phi(\tilde{\boldsymbol{\theta}}; \mathbf{x}_i) = h_i(\tilde{\mathbf{u}}^{(l)})$. If $l = m$ then we denote by $h_i$ the function $\mathbf{u}^{(m)} \mapsto W^{(m)} \mathbf{x}_i^{(m-1)}$, thus we also have $\Phi(\tilde{\boldsymbol{\theta}}; \mathbf{x}_i) = h_i(\tilde{\mathbf{u}}^{(m)})$.

Since $\tilde{\boldsymbol{\theta}}$ is a KKT point of Problem 1, then there are $\lambda_1, \ldots, \lambda_n$ such that

$$\tilde{\mathbf{u}}^{(l)} = \sum_{i \in [n]} \lambda_i \frac{\partial \left(y_i \Phi(\tilde{\boldsymbol{\theta}}; \mathbf{x}_i)\right)}{\partial \mathbf{u}^{(l)}} = \sum_{i \in [n]} \lambda_i \frac{\partial}{\partial \mathbf{u}^{(l)}} \left[y_i \cdot h_i(\tilde{\mathbf{u}}^{(l)})\right] \;,$$

where $\lambda_i \geq 0$ for all $i$, and $\lambda_i = 0$ if $y_i \cdot h_i(\tilde{\mathbf{u}}^{(l)}) \neq 1$. Note that since the inputs to all neurons in layers $l, \ldots, m - 1$ in the computation $\Phi(\tilde{\boldsymbol{\theta}}; \mathbf{x}_i)$ are non-zero, then the function $h_i$ is differentiable at $\tilde{\mathbf{u}}^{(l)}$. Therefore in the above KKT condition we use the derivative rather than the Clarke subdifferential. Moreover, if the constraints in Problem 1 are satisfies in $\tilde{\boldsymbol{\theta}}$, then the constrains in Problem 3 are also satisfied in $\tilde{\mathbf{u}}^{(l)}$ w.r.t. $\tilde{\boldsymbol{\theta}}$. Hence, the KKT condition of Problem 3 w.r.t. $\tilde{\boldsymbol{\theta}}$ holds.

Also, note that since the inputs to all neurons in layers $l, \ldots, m - 1$ in the computation $\Phi(\tilde{\boldsymbol{\theta}}; \mathbf{x}_i)$ are non-zero, then the function $h_i$ is locally linear near $\tilde{\mathbf{u}}^{(l)}$. We denote this linear function by $\tilde{h}_i$. Therefore, $\tilde{\mathbf{u}}^{(l)}$ is a KKT point of the following problem

$$\min_{\mathbf{u}^{(l)}} \frac{1}{2} \left\|\mathbf{u}^{(l)}\right\|^2 \quad \text{s.t.} \quad \forall i \in [n] \quad y_i \tilde{h}_i(\mathbf{u}^{(l)}) \geq 1 \;.$$

Since the constrains here are affine and the objective is convex, then $\tilde{\mathbf{u}}^{(l)}$ is a global optimum of the above problem. Thus, there is a small ball near $\tilde{\mathbf{u}}^{(l)}$ where $\tilde{\mathbf{u}}^{(l)}$ is the optimum of Problem 3 w.r.t. $\tilde{\boldsymbol{\theta}}$, namely, it is a local optimum.

Finally, note that in the proof of Lemma C.6 the parameters vector $\boldsymbol{\theta}'$ is obtained from $\tilde{\boldsymbol{\theta}}$ by changing only the first layer. Hence, in ReLU networks GF might converge in direction to a KKT point of Problem 1 which is not a global optimum of Problem 3, even if all inputs to neurons are non-zero.

## C.12 PROOF OF THM. 6.1

We have

$$\frac{du}{dt} = -\frac{\partial \ell(y\Phi(\boldsymbol{\theta}; x))}{\partial u} = -\frac{\partial \ell(vw + u)}{\partial u} = -\ell'(vw + u) \ .$$

$$\frac{dw}{dt} = -\frac{\partial \ell(y\Phi(\boldsymbol{\theta}; x))}{\partial w} = -\frac{\partial \ell(vw + u)}{\partial w} = -\ell'(vw + u) \cdot v \ .$$

$$\frac{dv}{dt} = -\frac{\partial \ell(y\Phi(\boldsymbol{\theta}; x))}{\partial w} = -\frac{\partial \ell(vw + u)}{\partial v} = -\ell'(vw + u) \cdot w \ .$$

Since we also have $v(0) = w(0)$ then for every $t \geq 0$ we have $v(t) = w(t)$. Note that $\ell'(vw+u) < 0$ and hence all parameters $u, w, v$ are strictly increasing. Since $v(0) = w(0) > 1$ then for every $t \geq 0$ we have $v(t) = w(t) > 1$ and hence $\frac{du(t)}{dt} < \frac{dw(t)}{dt} = \frac{dv(t)}{dt}$. Therefore $u(t) \leq v(t) = w(t)$ for all $t$.

We now calculate the derivative of

$$\bar{\gamma}(t) = \frac{v}{\|\boldsymbol{\theta}\|} \cdot \frac{w}{\|\boldsymbol{\theta}\|} + \frac{u}{\|\boldsymbol{\theta}\|} = \frac{vw}{v^2 + w^2 + u^2} + \frac{u}{\sqrt{v^2 + w^2 + u^2}} \ .$$

We have

$$\frac{d}{dt} \left[ \frac{vw}{v^2 + w^2 + u^2} \right] = \frac{1}{(v^2 + w^2 + u^2)^2} \left[ \left( -\ell'(vw + u)w^2 - \ell'(vw + u)v^2 \right) (v^2 + w^2 + u^2) \right.$$
$$\left. -vw \left( 2v(-\ell'(vw + u)w) + 2w(-\ell'(vw + u)v) + 2u(-\ell'(vw + u)) \right) \right] \ ,$$

and by plugging in $w = v$ the above equals

$$\frac{1}{(2v^2 + u^2)^2} \left[ \left( -2\ell'(v^2 + u)v^2 \right) (2v^2 + u^2) + v^2\ell'(v^2 + u) \left( 4v^2 + 2u \right) \right]$$
$$= \frac{-2\ell'(v^2 + u)v^2}{(2v^2 + u^2)^2} \left[ (2v^2 + u^2) - \left( 2v^2 + u \right) \right] = \frac{-2\ell'(v^2 + u)v^2 u(u - 1)}{(2v^2 + u^2)^2} \ .$$

Next, we have

$$\frac{d}{dt} \left[ \frac{u}{\sqrt{v^2 + w^2 + u^2}} \right] = \frac{1}{v^2 + w^2 + u^2} \left[ (-\ell'(vw + u)) \sqrt{v^2 + w^2 + u^2} \right.$$
$$\left. -u \frac{1}{2\sqrt{v^2 + w^2 + u^2}} \left( 2v(-\ell'(vw + u)w) + 2w(-\ell'(vw + u)v) + 2u(-\ell'(vw + u)) \right) \right]$$
$$= \frac{-\ell'(vw + u)}{v^2 + w^2 + u^2} \left[ \sqrt{v^2 + w^2 + u^2} - u \frac{1}{2\sqrt{v^2 + w^2 + u^2}} \left( 4vw + 2u \right) \right] \ ,$$

and by plugging in $w = v$ the above equals

$$\frac{-\ell'(v^2 + u)}{2v^2 + u^2} \left[ \sqrt{2v^2 + u^2} - u \frac{1}{2\sqrt{2v^2 + u^2}} \left( 4v^2 + 2u \right) \right]$$
$$= \frac{-\ell'(v^2 + u)}{2v^2 + u^2} \left[ \frac{2v^2 + u^2 - u \left( 2v^2 + u \right)}{\sqrt{2v^2 + u^2}} \right] = \frac{2\ell'(v^2 + u)v^2(u - 1)}{2v^2 + u^2} \cdot \frac{1}{\sqrt{2v^2 + u^2}} \ .$$

Overall, we have

$$\frac{d\bar{\gamma}(\boldsymbol{\theta})}{dt} = \frac{-2\ell'(v^2 + u)v^2 u(u - 1)}{(2v^2 + u^2)^2} + \frac{2\ell'(v^2 + u)v^2(u - 1)}{2v^2 + u^2} \cdot \frac{1}{\sqrt{2v^2 + u^2}}$$
$$= \frac{-2\ell'(v^2 + u)v^2(u - 1)}{2v^2 + u^2} \left[ \frac{u}{2v^2 + u^2} - \frac{1}{\sqrt{2v^2 + u^2}} \right]$$
$$= \frac{-2\ell'(v^2 + u)v^2(u - 1)}{2v^2 + u^2} \left[ \frac{u - \sqrt{2v^2 + u^2}}{2v^2 + u^2} \right] < 0 \ ,$$

where the inequality is since $u > 1$ and $v > 0$.

We now show that $\lim_{t\to\infty} \mathcal{L}(\boldsymbol{\theta}(t)) = 0$. Assume that $\|\boldsymbol{\theta}\| \not\to \infty$. Let $M > 0$ be such that $vw + u \le M$ for all $t$. Then, $-\ell'(vw + u) \ge C$ for some constant $C > 0$. Thus, $\frac{du}{dt} \ge C$ for all $t$, which implies that $u \to \infty$ in contradiction to our assumption. Hence, we must have $\|\boldsymbol{\theta}\| \to \infty$. Since $v = w \ge u \ge 1$ for all $t$, then we have $v \to \infty$ and $w \to \infty$. Therefore $\lim_{t\to\infty} \mathcal{L}(\boldsymbol{\theta}(t)) = \lim_{t\to\infty} \ell(vw + u) = 0$.

It is easy to verify that $\bar{\gamma}(\boldsymbol{\theta}(0)) = \frac{4}{12} + \frac{2}{\sqrt{12}} > 0.9$. In order to obtain $\lim_{t\to\infty} \bar{\gamma}(\boldsymbol{\theta}(t)) = \frac{1}{2}$ we first show that $\lim_{t\to\infty} \frac{u}{v} = 0$. Let $M > 4$ be some large constant. We show that there exists some $t'$ such that $\frac{v(t)}{u(t)} \ge M$ for all $t \ge t'$. Since $v \to \infty$, then there is some $t_1$ such that $v(t_1) = 2M$, and $\Delta > 0$ such that $v(t_1 + \Delta) = 2M + M^3$. Since $u \le v$ then $u(t_1) \le 2M$. For every $t \ge t_1$ we have $\frac{dv(t)}{dt} = -\ell'(vw + u) \cdot v \ge -\ell'(vw + u) \cdot 2M$. Also, we have $\frac{du(t)}{dt} = -\ell'(vw + u) \le \frac{1}{2M} \cdot \frac{dv(t)}{dt}$. Hence, we have

$$u(t_1 + \Delta) = u(t_1) + \int_{t_1}^{t_1+\Delta} \frac{du(t)}{dt} dt \le 2M + \int_{t_1}^{t_1+\Delta} \frac{1}{2M} \frac{dv(t)}{dt} dt = 2M + \frac{1}{2M}\left(v(t_1 + \Delta) - v(t_1)\right)$$

$$= 2M + \frac{1}{2M}\left(2M + M^3 - 2M\right) = 2M + \frac{M^2}{2} \le M^2 .$$

Thus,

$$\frac{v(t_1 + \Delta)}{u(t_1 + \Delta)} \ge \frac{2M + M^3}{M^2} \ge M .$$

Denote $t' = t_1 + \Delta$. Note that for every $t \ge t'$ we have

$$v(t) = v(t') + \int_{t'}^{t} \frac{dv}{dt} dt = v(t') + \int_{t'}^{t} (-\ell'(vw + u))w\,dt = v(t') + \int_{t'}^{t} (-\ell'(vw + u))v\,dt$$

$$\ge v(t') + \int_{t'}^{t} (-\ell'(vw + u))M^3\,dt = v(t') + M^3 \int_{t'}^{t} \frac{du}{dt} dt$$

$$\ge M \cdot u(t') + M^3\left(u(t) - u(t')\right) \ge M \cdot u(t') + M\left(u(t) - u(t')\right) = Mu(t) .$$

We have

$$\bar{\gamma}(\boldsymbol{\theta}(t)) = \frac{vw}{v^2 + w^2 + u^2} + \frac{u}{\sqrt{v^2 + w^2 + u^2}} = \frac{v^2}{2v^2 + u^2} + \frac{u}{\sqrt{2v^2 + u^2}} .$$

Since $\lim_{t\to\infty} \frac{u}{v} = 0$ then $\lim_{t\to\infty} \frac{u}{\sqrt{2v^2+u^2}} = 0$ and $\lim_{t\to\infty} \frac{v^2}{2v^2+u^2} = \frac{1}{2}$. Hence, $\lim_{t\to\infty} \bar{\gamma}(\boldsymbol{\theta}(t)) = \frac{1}{2}$ as required. Finally, since $\lim_{t\to\infty} \frac{u}{v} = 0$ and $w(t) = v(t) > 0$ for all $t$, then $\frac{\boldsymbol{\theta}}{\|\boldsymbol{\theta}\|}$ converges to $\left[\frac{1}{\sqrt{2}}, \frac{1}{\sqrt{2}}, 0\right]$. Thus, $\boldsymbol{\theta}$ converges in direction.

