# OpenReview forum: "On Margin Maximization in Linear and ReLU Networks"
_ICLR.cc/2022/Conference — ICLR 2022 Submitted_

### Official Review · Reviewer_frEy · 2021-11-02

**Correctness:** 4
**Technical Novelty And Significance:** 3
**Empirical Novelty And Significance:** Not applicable
**Recommendation:** 6
**Confidence:** 4

**Main Review:**

The topic of this paper is interesting, I think that these margin maximization questions are very important. But it suffers from being a collection of a large amount of small results. The paper goes over a large quantity of different settings, and it can be a bit overwhelming and difficult to follow.

Nevertheless the paper is a good and quite thorough overview of what happens in all these settings (though I would have been interested to study what happens for multiple outputs as this could make things more interesting in the fully-connected case for example). The counterexample are well described and quite simple, giving an idea of what structure leads to KKT points begin non-optimal. The authors mention that some of their counter example are stable under perturbation while some aren't, but don't really go in the details. I think that this is an important question. To make an analogy, the loss surface of shallow linear network has multiple saddles but the probability to converge to them is 0 since they are all strict saddles. I think it is quite likely that a similar phenomenon could happen here where some of the counterexamples presented in this article only happen with probability zero or a very small probability.

I found the non-homogeneous example interesting, though it is very simple.

**Summary Of The Paper:**

This paper is inspired by recent results which have shown that homogeneous networks, when trained on losses with exponential decay converge in direction to KKT points, i.e. points which are first order stationary w.r.t. the opimization of the margin of classification. Since this margin maximization is non-convex, KKT points may not be global maximizers of the margin or not even local maximizers. The authors therefore study a wide range of architecture of linear and ReLU networks, some shallow, some deep, with or without sparse and shared weights, and study in which case all KKT points are global margin maximizers, local margin maximizers or neither (for negative results, counterexamples are given).

Finally they study what happens for non-homogeneous networks, in particular in the presence of skip connection. While for homogeneous networks the margin increases during training, they give an example of a simple network with skip connection where the margin is strictly decreasing.

**Summary Of The Review:**

The paper studies an important question and is quite thorough. However it suffers from being a collection of many rather small results.

---

> ### Author Response · Authors · 2021-11-22
> **Response**
>
> We thank the reviewer. Below we address the main comments:
>
> “it suffers from being a collection of a large amount of small results. The paper goes over a large quantity of different settings, and it can be a bit overwhelming and difficult to follow”:
> The paper considers many different settings and characterizes the guarantees on margin maximization in each setting. Accordingly, it draws a somewhat complicated picture. We summarized the main results in two tables in the introduction, but we agree that having many different settings and results makes it a bit overwhelming. Nevertheless, we believe that the paper gives a rich characterization of when gradient flow maximizes the margin. We believe that this problem is crucial for understanding the implicit bias in neural networks. We will add a short discussion section in order to make the main message clearer.

---

### Official Review · Reviewer_7GJf · 2021-11-03

**Correctness:** 4
**Technical Novelty And Significance:** 3
**Empirical Novelty And Significance:** Not applicable
**Recommendation:** 6
**Confidence:** 4

**Main Review:**

#### Strengths

1. This paper presents lots of positive and negative results on the implicit bias of gradient flow, which certainly increases our knowledge of when gradient flow can or cannot converge to a local/global max-margin solution.
2. Among the positive results, the most interesting one is the per-layer local optimality for deep ReLU nets. Given the negative results in this paper, the per-layer local optimality seems to be the last hope for providing theoretical guarantees for margin in deep learning. Also this per-layer local optimality could have potential consequences besides margin.
3. The counterexamples for margin maximization are simple and intuitive.

#### Weaknesses

1. Although the authors made a lot of efforts analyzing the margin maximization across various different settings, this paper lacks a key result to highlight, and thus it may not be clear to readers what the main message is. Also this paper does not contain a conclusion section. The results for the linear nets reconstruct previous results with slight improvement, and the results for the ReLU nets are mostly negative. In my point of view, the result that interests me most is the per-layer local optimality for deep ReLU nets, and the main message could also be "proving the global optimality of margin is hopeless".
2. Although I like the simplicity of the counterexamples, I could not get a high-level insight and do not understand when the counterexamples may conceptually hold. Most of the counterexamples look like corner cases that could be excluded easily:
   * Theorem 3.2 considers a 2-neuron net. If the width is slightly wider, say 10, then we may get the max margin since there are more weight vectors with $\langle w, x_i \rangle > 0$.
   * Theorem 4.1, 4.4, 5.1 use symmetry: some parameters are initially the same and there is no symmetry breaking during training. So random initialization may mitigate this issue.
3. The relationship between this paper and Chizat & Bach (2020), Ji & Telgarsky (2020) needs more explanations. Specifically, these two previous works analyzed the neural net with infinite or exponential width. Ji & Telgarsky (2020) actually showed that the global optimality of margin can be attained if a covering condition holds: the directions of weight vectors in the first layer forms an $\epsilon$-covers of the unit sphere. This means any counterexample should describe a training trajectory that violates the covering condition somehow. I would like to see if the authors can give high-level insights into why the covering conditions are violated.
4. The counterexample on the non-homogeneous neural net does not seem to be very interesting. Note that the normalized margin itself is not well-defined for non-homogeneous neural nets, because dividing by norms does not make perfect sense anymore. Although Lyu & Li (2020) noticed that the monotonicity of normalized margin (defined by dividing the product of weight norms) for VGG with bias, to actually analyze this a better definition for the normalization margin is still needed. Thus it would be more interesting to have positive results with a well-defined normalized margin, or negative results against a broad class of normalized margin definitions.

#### Minor Comments

* The footnote 3 could be misleading. A possible misinterpretation could be: Ji & Telgarsky (2020) only proved margin maximization in the predictor space, but the current paper proves that in the parameter space with brand new techniques. This is not true because the proof by Ji & Telgarsky (2020) almost implies margin maximization in the parameter space, and this paper indeed used Proposition 4.4 from Ji & Telgarsky (2020).
* Although the proofs for Theorem 5.2 and 5.4 are simple enough now, I am wondering whether it can be further simplified by applying Corollary 4.5 from Lyu & Li (2020). More specifically, for any homogeneous neural net, we can write $\Phi(\theta; x_i)$ as $\langle \theta, \nabla \Phi(\theta; x_i) \rangle$. Then we change the weight of the i-th layer (which corresponds to a part of $\theta$) while fixing the others. For deep linear nets, we can see from the formula that $\Phi(\theta; x_i)$ is linear with the i-th layer weight; for deep ReLU nets, $\Phi(\theta; x_i)$ is locally linear with the layer weight. Lyu & Li (2020) showed the KKT convergence result for the original net. Fixing the weights of other layers implies the per-layer global optimality for linear nets. Since the KKT conditions are local conditions, the KKT conditions still hold if a ReLU net is linear everywhere wrt the i-th layer, which implies the per-layer global optimality for linear nets (as what this paper and Lyu & Li (2020) did). I believe this proof also shows how to generalize Theorem 5.2 and 5.4 to homogeneous nets besides ReLU and linear nets.
* Section 2, Optimization problem and gradient flow (GF): it should be clarified that subgradient flow in the sense of Clarke is used for ReLU nets.

**Summary Of The Paper:**

This paper studies the implicit bias of gradient flow in the lens of margin maximization. A previous work by Lyu & Li (2020) shows that gradient flow directionally converges to a KKT point of the margin maximization problem. The authors then focus on understanding what kind of local/global optimality of margin can be guaranteed by applying the KKT convergence result. The authors consider various settings on 2-layer/deep linear/ReLU neural nets, and for each setting they prove either local/global optimality of margin or an impossibility result.

As the positive results could be more preferable by the ML community, I summarize the positive results below:

* Global optimality for (deep) fully-connected linear nets;
* Global optimality for general two-layer linear nets with no weight sharing, assuming all the weights are non-zero;
* Local optimality for general two-layer ReLU nets with no weight sharing, assuming all neurons have non-zero inputs;
* Per-layer global optimality for general (deep) linear nets (i.e., every layer weights achieve the global max margin if all other layers are fixed.)
* Per-layer local optimality for general (deep) ReLU nets (i.e., every layer weights achieve a local max margin if all other layers are fixed.), assuming all neurons have non-zero inputs.

Here "general" nets can have arbitrary connections between neurons in adjacent layers.

**Summary Of The Review:**

This paper studies an interesting topic and presents a lot of positive and negative results, which could be potentially used by other theoretical works later. But this paper lucks a key result to highlight. I appreciate the authors' efforts in analyzing the various settings in the paper, but I have to vote for weak rejection because of the lack of a key result and all the other weaknesses I mentioned above.

-----------------

I was mainly concerned that this paper lacks a key result to highlight, but after reading the rebuttal, I am now satisfied with the authors' response to this major concern. While my other concerns remain (and the authors did not respond to most of them), I am happy to increase my score to 6, assuming that the authors will add the short conclusion section as promised. I look forward to the next revision of the paper.

---

> ### Author Response · Authors · 2021-11-22
> **Response**
>
> We thank the reviewer for the comments, and will fix the minor issues. Below we address the main comments.
>
> “this paper lacks a key result to highlight, and thus it may not be clear to readers what the main message is”:
> The paper considers many different settings and characterizes the guarantees on margin maximization in each setting. Accordingly, it draws a somewhat complicated picture. We summarized the main results in two tables in the introduction, but we agree that having many different settings and results makes it harder to phrase a simple message. Our negative results suggest that even in very simple settings gradient flow does not maximize the margin even locally, and we believe that these negative results should be used as a starting point for studying which assumptions are required for proving margin maximization. We believe that the negative results imply that in order to better understand the implicit bias we need a more careful analysis which includes assumptions on the dataset and on the initialization of gradient flow. Our positive results take a first step in that direction, and show that under some assumptions, which we believe to be reasonable in realistic settings, gradient flow maximizes the margin (either locally or globally). Also, the notion of per-layer margin maximization suggests another path for obtaining positive results on the implicit bias.
> Overall, the answer to “when does gradient flow maximize the margin?” is complicated, but we believe that the paper makes a large step towards a better understanding of the problem.
> We will add to the paper a short conclusion section.
>
> “Most of the counterexamples look like corner cases that could be excluded easily”:
> As often the case with negative results, they show that without additional assumptions the claims do not hold. In our case, the negative results imply that in order to obtain guarantees for margin maximization (even locally) we must add assumptions that circumvent the negative results. For example, as you suggested, a possible direction for circumventing our negative result on fully-connected ReLU networks is to analyze wide depth-$2$ ReLU networks (with random initialization).
>
> “The relationship between this paper and Chizat & Bach (2020), Ji & Telgarsky (2020) needs more explanations”:
> We will add a discussion on the relation to these papers.

---

### Official Review · Reviewer_sBmy · 2021-11-04

**Correctness:** 4
**Technical Novelty And Significance:** 3
**Empirical Novelty And Significance:** Not applicable
**Recommendation:** 6
**Confidence:** 3

**Main Review:**

Strengths:
- Findings are interesting and concern an important topic
- Paper is generally well-written and rather easy to follow
- main findings are conveniently summarized on a Table

Weaknesses:
- The paper is missing a sound conclusion/discussion on the implications of their findings.

Given the interesting negative findings of the paper, I feel it is missing a punchline: what is the implication of these findings about existing works on implicit bias? What is the step ahead from the authors' viewpoint? Suppose that global optimality was true more generally, what are the implications of this? How could this be useful for further theoretical analysis? And in cases where is not true, what does this imply about works investigating implicit bias?

Related to above: in view of your Thm. 5.2, do you believe that is possible more generally to characterize the converging point as global optimum of some other problem with KKT conditions a superset of the KKT conditions of the max margin Problem 1?

- In the abstract it is stated that: "On the flip side, we identify multiple setting where a local/global optimum can be guaranteed". This reads very promising compared to the sole positive results of Thm 3.1 and Thm 5.2 (Thm 4.2a requires rather stringent non-zero weight conditions)

Questions to the authors:
- The paper discusses binary classification. Have the authors considered what can be said about multiclass classification?
- Is program (3) equivalent to min_\theta \max_\ell ||u^{\ell}||_2  s.t. y_i\Phi(\theta;x_i)\geq 1 ? That is, does this imply that GF converges to the program that maximizes the worst margin among all layers?
- In Remark 4.1: (2) is a convex program, so every local optimum is a global optimum. "linear predictor \tilde\beta is not a local optimum of the following program" might be confusing
- Finally, can the authors comment explicitly on the role of bias?

**Summary Of The Paper:**

The paper studies implicit bias of linear and Relu networks (fully connected, diagonal, convolutional etc). Specifically, it investigates deeper a result by Lyu & Li (2019) who showed that gradient flow of such homogeneous next converges to a KKT point of the max margin problem in the parameter space. Owing to its non-convex nature, KKT conditions of this max margin problem do not guarantee local/global optimality. The paper answers whether these KKT conditions are (or not) local/global optima for a variety of linear and homogeneous Relu networks.

The results are perhaps "negative" in the sense that Global optimality is found to be true only for Fully connected Linear (deep) nets, unless additional assumptions are made (Thm 3.1). An interesting positive result is that for linear deep nets not necessarily fully connected (e.g. diagonal/convolutional) the converging point of GF is a global optimum of a program that maximizes the margin of each layer independently.

**Summary Of The Review:**

The paper has interesting findings on implicit bias theory. While the findings themselves are presented in a rather clear manner, the paper is lacking a sound conclusion especially so since it is "challenging" existing beliefs. I would encourage the authors to attempt clarifying their view on how these results are expected to affect further research on implicit bias.

---

> ### Author Response · Authors · 2021-11-22
> **Response**
>
> We thank the reviewer for the comments. Below we address the main questions that have been raised.
>
> “I feel it is missing a punchline: what is the implication of these findings about existing works on implicit bias? What is the step ahead from the authors' viewpoint? Suppose that global optimality was true more generally, what are the implications of this? How could this be useful for further theoretical analysis? And in cases where is not true, what does this imply about works investigating implicit bias?”:
> First, the paper considers many different settings and characterizes the guarantees on margin maximization in each setting. Accordingly, it draws a somewhat complicated picture. We summarized the main results in two tables in the introduction, but we agree that having many different settings and results makes it harder to phrase a simple punchline. We believe that understanding margin maximization is crucial for explaining generalization in deep learning, which is the main motivation for studying the implicit bias. Thus, if global optimality was true more generally, then standard norm-based generalization bounds for neural networks might be able to explain generalization in deep learning, at least partially. Our negative results suggest that even in very simple settings gradient flow does not maximize the margin even locally, and we believe that these negative results should be used as a starting point for studying which assumptions are required for proving margin maximization. We believe that the negative results imply that in order to better understand the implicit bias we need a more careful analysis which includes assumptions on the dataset and on the initialization of gradient flow. Our positive results take a first step in that direction, and show that under some assumptions, which we believe to be reasonable in realistic settings, gradient flow maximizes the margin. Also, the notion of per-layer margin maximization suggests another path for obtaining positive results on the implicit bias. Overall, the answer to “when does gradient flow maximize the margin?” is complicated, but we believe that the paper makes a large step towards a better understanding of the problem.
> We will add to the paper a short discussion on the above issues .
>
> “in view of your Thm. 5.2, do you believe that is possible more generally to characterize the converging point as global optimum of some other problem with KKT conditions a superset of the KKT conditions of the max margin Problem 1?”:
> This is a good question. Thm 5.2 indeed suggests that in order to obtain global optimality we should change the problem that we consider. Namely, instead of considering margin maximization of all layers simultaneously, Thm 5.2 considers margin maximization for each layer separately. Perhaps this approach might lead to more results. Right now we don’t have a concrete idea.
>
> ‘In the abstract it is stated that: "On the flip side, we identify multiple setting where a local/global optimum can be guaranteed". This reads very promising compared to the sole positive results of Thm 3.1 and Thm 5.2 (Thm 4.2a requires rather stringent non-zero weight conditions)’:
> First, we argue that when training depth-2 neural networks in practice, gradient descent often converges to networks that do not contain zero neurons. Hence, we believe that the assumption in Thm 4.2a is realistic. Second, note that in addition to the theorems that you mentioned, there are also some positive results for local optimality in Theorems 4.3 and 5.4.
>
> “The paper discusses binary classification. Have the authors considered what can be said about multiclass classification?”:
> This is a good question. Since the paper already considers many different settings we felt that extending it further might make it more confusing / overwhelming for the reader. Hence, we left the extension to the multiclass setting for future work. We also left other possible extensions for future work, e.g., considering other activations such as leaky-ReLU.
>
> “Does this imply that GF converges to the program that maximizes the worst margin among all layers?”
> We believe that the answer is negative, but don’t have a counterexample.
>
> ‘In Remark 4.1: (2) is a convex program, so every local optimum is a global optimum. "linear predictor \tilde\beta is not a local optimum of the following program" might be confusing’:
> Right. We will remove the word “local” from this sentence.
>
> “Finally, can the authors comment explicitly on the role of bias?”
> We assume that you mean neural networks where the neurons have bias terms. Since we are interested here in homogeneous networks, then bias terms can be considered only in the first hidden layer (otherwise the network is not homogeneous). This is an interesting question, but we decided to leave this setting for future research.

---

> > ### Comment · Reviewer_sBmy · 2021-12-06
> > **Response**
> >
> > Thank you for the responses.
> >
> > I think the paper has some interesting findings on an important topic.
> > In line with their response above, I hope that the authors consider including a discussion on broader implications of the negative results and possible avenues to address them, as this will likely further benefit the community.
> >
> > A couple of remaining 'concerns' (which lead me to keeping my score at 6):
> > - it is somewhat unclear in the current form if the individual proofs have a distilled "proof technique" that can be useful in more general settings.
> > - If I understand it correctly, the "positive" finding of Thm 5.2 (aka Problem 3) is nothing but a reinterpretation of convergence to KKT points of Problem 1. It is not clear to me why this interpretation is useful, e.g. are there settings in which it is possible to explicitly characterize the global optima of Problem 3?

---

> > > ### Author Response · Authors · 2021-12-08
> > > **Response**
> > >
> > > Thanks.
> > >
> > > In the conclusion section we will include a discussion on the possible ways to circumvent our negative results and on their implications.
> > >
> > > Regarding the proof techniques: First, in our positive results on depth-2 networks (Theorems 4.2 and 4.3), we show that the solution is a KKT point not only for the max-margin problem in parameter space, but also for another convex problem whose objective is of the form $\sum_i \lVert u_i \rVert$. It allows us to show global/local optimality. We believe that this method might be relevant also for other problems on depth-2 networks. Second, we believe that some ideas from our negative results might be helpful to obtain negative results for convergence to a global/local max-margin (or min-norm) solutions also in other settings.
> > >
> > > Regarding Thm 5.2: Problem (3), namely, per-layer margin maximization, is a weaker notion of margin maximization where global/local optimality is guaranteed. It means that even if GF fails to find an optimum for Problem (1), it still maximizes the margin for each layer separately. In other words, if GF converges to a “bad” solution, then it must be a solution that maximizes the margin in each layer. The known generalization bounds for neural networks are based on the norms of all layers, and hence the per-layer margin maximization does not give immediate consequences in terms of generalization (as far as we can see). However, it seems like a natural property of the implicit bias of deep networks, which might be helpful in future research.

---

### Official Review · Reviewer_AttS · 2021-11-06

**Correctness:** 4
**Technical Novelty And Significance:** 2
**Empirical Novelty And Significance:** Not applicable
**Recommendation:** 5
**Confidence:** 2

**Main Review:**

There are some concerns about the technical contributions of the paper. First, the implications of the technical result are not clear, in particular for generalization ability. In other words, how the characterization affect the generalization performance of the network?

The second concern is that it is not clear how significant the generalization/improvement of the results is, compared to the result of Lyu & Li (2019) .  So far, it looks to me that the results are not as significant as the previous work.

**Summary Of The Paper:**

The paper investigates characterizations of solutions for the problem of training neural networks with exponential loss or logistic loss. The main contribution of the paper is to extend the previous work of  Lyu & Li (2019) for various types of networks.

**Summary Of The Review:**

The technical results seems non-trivial but the implications are not clear.

---

> ### Author Response · Authors · 2021-11-22
> **Response**
>
> We thank the reviewer for the comments, which we address below.
>
> Regarding the implications on generalization: Margin maximization is equivalent to norm minimization if we normalize the network weights such that it attains margin $1$. There are many works that show the implications of small weights on generalization. Also, explicit regularization is common in practice for this reason. In recent years there have been many works on the implicit bias in neural networks. The reason that the study of implicit bias has attracted so much interest, is that by understanding the implicit bias we might be able to better understand generalization in neural networks. Hence, it is important to understand whether margin maximization occurs when training neural networks with gradient based methods. In this work we show both negative and positive results, which improve our understanding of this issue.
>
> “it is not clear how significant the generalization/improvement of the results is, compared to the result of Lyu & Li (2019) . So far, it looks to me that the results are not as significant as the previous work”:
> In Lyu & LI (2019) it was shown that gradient descent/flow converges to a KKT point of the max-margin problem. However, it does not guarantee that the margin is actually maximized, neither globally or locally. Since understanding this issue is a fundamental problem in the study of implicit bias, the aim of our work is to characterize when gradient flow maximizes the margin. Thus, our results complement the result from Lyu & Li (2019). The result of Lyu & Li (2019) is fundamental and has very high impact, so we disagree with the approach that it should be considered as the lower threshold for evaluating whether a paper should be published in ICLR (or any other venue).

---

### Decision · Program_Chairs · 2022-01-20

**Decision:**

Reject

**Comment:**

This paper studies margin maximization in linear and ReLu networks. The reviewers appreciate the technical contributions of this paper, especially the simple counterexamples. However, reviewers also found the new results seem not to give enough conceptual insights or an important "main result". The meta reviewer agrees and thus decides to reject this paper.